# SimGFM: Simplifying Discrete Flow Matching for Graph Generation

**Chunyu Luo**[* 3]  **Yuankai Luo**[* ✉ 1 2]  **Xiao-Ming Wu**[4]  **Lei Shi**[✉ 3]

## Abstract

Discrete Flow Matching (DFM) presents a promising approach for graph generation; however, existing adaptations often introduce substantial complexity by incorporating task-specific heuristics, compromising the continuity equation and significantly expanding the hyperparameter space. Moreover, their sampling efficiency remains limited, as the required number of steps is often comparable to diffusion models, diminishing DFM's practical advantages. To address these limitations, we propose SimGFM, a simplified graph DFM for graph generation. Leveraging characteristic patterns in graph-generation trajectories, SimGFM relies only on the scheduler and rate matrix, eliminating heavy heuristics and hyperparameter tuning, and achieves large step reductions while preserving SOTA results. SimGFM achieves strong empirical results: on QM9, it matches prior models requiring 500–1000 steps with only 10 steps, and on most datasets, its performance at 50 steps matches or surpasses these baselines, demonstrating both efficiency and competitiveness. Our source code is available at https://github.com/LUOyk1999/SimGFM.

## 1. Introduction

Graph generation is fundamental across domains from molecular chemistry to social networks, as graphs compactly represent complex relations and generate realistic structured data. Recent advances include continuous-time discrete diffusion frameworks (Xu et al., 2024; Siraudin et al., 2024) and discrete-flow frameworks (QIN et al., 2025; Campbell et al., 2024; Gat et al., 2024).

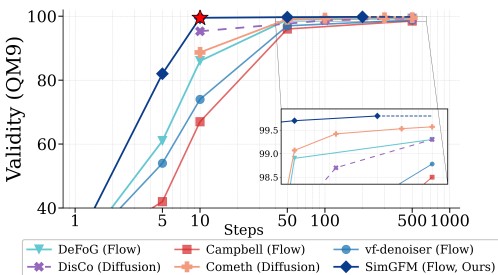

*Figure 1.* Validity on QM9 vs. sampling steps. Campbell (red) requires many steps, while vf-denoisers (blue) are more efficient. However, as current graph DFM methods still require sampling steps comparable to diffusion models, SimGFM significantly advances efficiency, reaching over 99% validity in 10 steps.

Diffusion models (Ho et al., 2020; Nichol & Dhariwal, 2021; Vignac et al., 2022) tightly couple training and sampling: once components such as the noise schedule or rate matrix are modified (Nichol & Dhariwal, 2021; Karras et al., 2022; Xu et al., 2024; Siraudin et al., 2024), retraining is typically required, incurring substantial computational cost. By contrast, discrete-flow models (Campbell et al., 2024; Gat et al., 2024) decouple training from sampling, allowing sampling adaptations without retraining and thus greater flexibility for diverse data distributions. In CV/NLP, flow matching has markedly accelerated sampling, in some cases enabling near one-step generation (Song et al., 2023; Liu et al., 2022; Lee et al., 2024; Geng et al., 2025). However, in graph generation, existing discrete-flow models remain computationally costly and require nearly as many steps as diffusion-based approaches, leaving the potential sampling efficiency of flow matching largely unrealized (QIN et al., 2025; Hou et al., 2025).

As shown in Fig. 1, Campbell et al. (2024) derive a closed-form rate matrix (Eq. 5) from the posterior endpoint $p_{1|t}(\cdot \mid X_t)$, but its posterior expectations and combinatorial bookkeeping are costly for graphs. Building on this, recent SOTA model (QIN et al., 2025) augments the Campbell field with heuristic velocity terms to gain accuracy, at the cost of (i) potential violations of the continuity equation, and (ii) added methodological complexity. By contrast, the vf-denoiser (Gat et al., 2024) offers a concise scheduler-based formulation (Eq. 6), avoids posterior expectations, and shows strong few-step performance, making it a simpler and more effective backbone for graph DFM.

[*]Equal contribution. Work done at Nanjing University. [1]State Key Laboratory of Novel Software Technology, Nanjing University [2]School of Artificial Intelligence, Nanjing University [3]Beihang University [4]The Hong Kong Polytechnic University. ✉ Correspondence to: Yuankai Luo <yuankailuo@nju.edu.cn>.

*Proceedings of the 43rd International Conference on Machine Learning*, Seoul, South Korea. PMLR 306, 2026. Copyright 2026 by the author(s).

Motivated by these observations, we propose SimGFM, a vf-denoiser-based method for graph generation that keeps the sampling dynamics strictly consistent with the DFM continuity equation, without introducing auxiliary modules. SimGFM makes three key design choices. First, we instantiate the executable rate matrix via the vf-denoiser formulation, which avoids costly posterior expectations (and Monte Carlo approximations) and enables an implementation-friendly sampler (Sec. 3.1). Second, guided by our trajectory-level observation that valid graph structures tend to emerge predominantly near the endpoint under uniform denoising, we adopt an endpoint-focused scheduler that allocates more updates as $t \to 1$ (Sec. 3.2), enabling denser refinement at the terminal stage. Third, for such endpoint-focused schedulers, we introduce *Endpoint Safety Projection* (ESP; Sec. 3.3) to ensure that the endpoint-window updates remain numerically valid.

On QM9 (Wu et al., 2018), SimGFM achieves 99.5% validity in just 10 steps, and across most datasets, it reaches or approaches SOTA performance with 10–50 steps, representing an *order-of-magnitude reduction* compared with diffusion/flow baselines (typically 500–1000), while also lowering hyperparameter tuning burden.

## 2. Preliminaries

### 2.1. Discrete Flow Matching

In this section, we introduce the core concepts of Discrete Flow Matching (DFM) (Campbell et al., 2024; Gat et al., 2024). Unlike diffusion models, which learn a data distribution via iterative noising and denoising, the goal of DFM is to learn a deterministic *probability path* $p_t$ from a simple source distribution $p_0$ (e.g., a sequence composed of a "mask" symbol) to a target data distribution $p_1$. The core of the model is to train a neural network to predict the *velocity field* $u_t$ of this probability path, which guides how samples evolve with time $t \in [0, 1]$ from the source to the target.

To build this framework, we first define a **conditional probability path** from a specific source sample $x_0 \sim p_0$ to a specific target sample $x_1 \sim p_1$. A simple and effective choice is their convex combination:

$$p_t\big(x^i \mid x_0, x_1\big) = (1 - \kappa_t)\, \delta_{x_0^i}\big(x^i\big) + \kappa_t\, \delta_{x_1^i}\big(x^i\big), \quad (1)$$

where $x^i$ is the $i$-th element of the sequence, $\delta$ is the Dirac delta (point mass), and $\kappa_t$ is a schedule increasing monotonically from $\kappa_0 = 0$ to $\kappa_1 = 1$. This formula states that at time $t = 0$, the sample coincides with the source $x_0$, and at $t = 1$ it fully transforms into the target $x_1$.

To simulate generation along the prescribed path $p_t(x)$ for $t \in [0, 1]$, DFM adopts the **continuous-time Markov chain** (CTMC) paradigm: the sample $X_t$ makes jumps over a state space $\mathcal{D}$ as time $t$ evolves continuously on $[0, 1]$. DFM

focuses on a model that predicts the **rate of change of probabilities** for each coordinate (token) of the current sample $X_t$ with $N$ tokens. Thus, for a sample $X_t \sim p_t$, each token updates independently as

$$X_{t+h}^i \ \sim \ \delta_{X_t^i}(\cdot) \ + \ h\, u_t^i(\cdot, X_t), \qquad (2)$$

where $\delta_{X_t^i}$ denotes a Dirac mass at the current value and $u_t^i$ is the probability velocity field for the $i$-th coordinate. If the probabilistic velocity $u_t$ *generates* the probability path $p_t$, it means that for all $t \in [0, 1)$ and any sample $x_t \sim p_t$, updating each position $i$ using the rule above (2) yields $x_{t+h} \sim p_{t+h} + o(h)$.

Moreover, the velocity $u_t$ should satisfy the following **rate conditions**:

$$\sum_{x^i \in [K]} u_t^i(x^i, z) = 0, \ u_t^i(x^i, z) \geq 0, \ \forall\, i \in [D], \ x^i \neq z^i.$$
$$(3)$$

Furthermore, prior work (Campbell et al., 2024; Gat et al., 2024) shows that a **continuity equation** (also called the Kolmogorov forward equation) holds in discrete flow matching, describing the time derivative of the state-marginal probability $\dot{p}_t(x), x \in \mathcal{S}$:

$$\dot{p}_t(x) \ + \ \mathrm{div}_x\big(p_t\, u_t\big) \ = \ 0, \qquad (4)$$

where $\mathrm{div}_x\big(p_t\, u_t\big) = \sum_{z \in \mathcal{S}} \sum_{i=1}^{D} \delta_x\big(z^{\bar{i}}\big) \Big[ p_t(x)\, u_t^i(x^i, x) - p_t(z)\, u_t^i(x^i, z) \Big]$, measures the total outflow (probability flow $x \to z$) minus total inflow ($z \to x$) at state $x \in \mathcal{S}$, and $\delta_x\big(z^{\bar{i}}\big) = \prod_{j \neq i} \delta_{x^j}(z^j)$ indicates that only pairs $(x, z)$ agreeing on all coordinates except possibly the $i$-th are considered when computing the flow. Intuitively, the continuity equation expresses that the rate of change of probability mass at $x$ equals the net effect of the probability flow $p_t u_t$ at $x$. It has been shown that if the continuity equation holds, then $u_t$ can generate the probability path $p_t$.

The choice of $u_t$ is crucial. Two commonly used constructions for the rate matrix are:

**(1) Campbell's construction.** (Campbell et al., 2024) provide a closed-form solution for the rate matrix $u_t$:

$$u_t^*(x, z | z_1) = \frac{\mathrm{ReLU}\left[\partial_t p_{t|1}(x \mid z_1) - \partial_t p_{t|1}(z \mid z_1)\right]}{Z_t^{>0} p_{t|1}(z \mid z_1)},$$
$$x \neq z. \qquad (5)$$

where $p_{t|1}(z \mid x)$ means the state $z$ at time $t$ given the state $x$ at time 1 and $Z_t^{>0} = \big|\{z_t : p_{t|1}(z_t \mid z_1) > 0\}\big|$, the diagonal case is set by $u_t^*(x, x | z_1) = -\sum_{x \neq z} u_t^*(x, z | z_1)$. Finally, the rate matrix is obtained by taking the posterior expectation: $u_t(x, z) = \mathbb{E}_{p_{1|t}(z_1 | z)}[u_t^*(x, z \mid z_1)]$.

**(2) Vf-denoiser.** (Gat et al., 2024) propose the vf-denoiser:

$$u_t^i(x^i, z) = \frac{\dot{\kappa}_t}{1 - \kappa_t} \left[ p_{1|t}\big(x^i \mid z\big) - \delta_{z^i}\big(x^i\big) \right], \quad (6)$$

where $p_{1|t}(x \mid z)$ means the state $x$ at time 1 given the state $z$ at time $t$ and $\kappa_t$ is a scheduler (a monotone time mapping) satisfying $\dot{\kappa}_t \geq 0$, $\kappa_0 = 0$, and $\kappa_1 = 1$.

Both constructions depend on the prior $p_{1|t}(\cdot \mid z_t)$, which is typically estimated by a trained model; we denote the model output by $p_{1|t}^\theta(\cdot \mid z_t)$. The training objective is

$$\mathcal{L}(\theta) = - \sum_{i \in [N]} \mathbb{E}_{t,\,(X_0, X_1),\,X_t} \log p_{1|t}^\theta \big(X_1^i \mid X_t\big). \quad (7)$$

## 2.2. Discrete Flow Matching on Graphs

Applying the Discrete Flow Matching (DFM) framework to graph generation requires accounting for the unique structure of graphs—namely, sets of nodes and edges. We represent a graph with $N$ nodes as $G = (X, E)$, where $X = \{x^{(i)}\}_{i=1}^N$ is the set of node attributes and $E = \{e^{(ij)}\}_{1 \leq i < j \leq N}$ is the set of edge attributes. Based on Eq. 1, the probability path over graphs factorizes as

$$p_t(G_t \mid G_0, G_1) = \prod_{i=1}^N p_t\Big(x_t^{(i)} \mid x_0^{(i)}, x_1^{(i)}\Big) \\ \prod_{1 \leq i < j \leq N} p_t\Big(e_t^{(ij)} \mid e_0^{(ij)}, e_1^{(ij)}\Big), \quad (8)$$

where $G_0 \sim p_0$ is a prior noise graph and $G_1 \sim p_1$ is a real data graph. Given this factorization, the sampling process for graphs follows the general update rule in Eq. 2: each node or edge is updated according to its velocity field,

$$G_{t+\Delta t}^{(k)} \sim \delta_{G_t^{(k)}}(\cdot) + \Delta t \cdot u_t^{(k)}(\cdot, G_t), \quad (9)$$

where $k$ denotes a node $(i)$ or an edge $(ij)$. Repeating from $t = 0$ to $t = 1$ generates a graph.

### 2.2.1. Existing Methods

Due to the complexity of graphs, graph generation is more challenging than image or text generation. Although DFM has solid theoretical foundations, directly applying it to complex graph structures often yields suboptimal results. Consequently, researchers have developed various auxiliary and heuristic techniques to enhance performance.

**Fine-tuning the model output.** This line of work optimizes the predictor $\phi_\theta$ to produce graphs with desired properties. For example, GGFLOW (Hou et al., 2025) adopts a two-stage strategy: first pretraining with standard flow-matching loss to learn $p_\theta(G_1 \mid G_t)$, and then fine-tuning via reinforcement learning (RL). Reward functions tied to graph

properties (e.g., docking scores, connectivity) guide RL, yielding an optimized policy $p_\theta^{\mathrm{RL}}(G_1 \mid G_t)$.

**Modifying the velocity field.** Another line directly alters the sampling dynamics. DEFOG (QIN et al., 2025), for instance, augments Campbell's base field (Eq. 5):

$$u_t(\cdot \mid G_1) = u_t^*(\cdot \mid G_1) + \omega\, u_t^\omega(\cdot \mid G_1) + \eta\, u_t^{\mathrm{DB}}(\cdot \mid G_1), \quad (10)$$

where $u_t^*$ is the base velocity from Campbell's construction, $u_t^\omega$ is a target-guidance term weighted by $\omega$, and $u_t^{\mathrm{DB}}$ is a stochastic exploration term weighted by $\eta$.

### 2.2.2. Challenges and Our Design Approach

**1. Violations of the continuity equation.** Directly fine-tuning the model output or modifying the velocity field (e.g., target-guidance heuristics) can break the core constraints required by Eq. 4, such as mass conservation and nonnegativity. In practice, these approaches often rely on auxiliary adjustments (e.g., normalization or clipping), which act as external interventions on the probability flow. While they may work empirically, such strategies lack a firm theoretical foundation and deviate from the standard DFM formulation.

**2. Methodological complexity.** Many enhancements to DFM introduce additional heuristics and design choices, increasing modeling complexity and reducing reproducibility. As shown in Figure 7, our experiments further indicate that, on some benchmarks, the SOTA variant (QIN et al., 2025) does not consistently outperform pure baselines.

**3. Sampling efficiency.** Flow models are valued for reducing the number of sampling steps compared to diffusion models. However, in graph generation, the steps required by current DFM methods remain comparable to those of diffusion approaches. This can be observed in Figure 1, where existing graph DFM methods require nearly the same number of steps as diffusion-based models, suggesting that the efficiency advantage of DFM is not yet fully realized.

**Our approach.** We adopt a minimalist design fully consistent with DFM theory (Sec. 3): a concise vf-denoiser-based executable rate that avoids posterior expectations and heavy heuristics to address Challenge 2 (Sec. 3.1); an endpoint-focused scheduler that improves few-step efficiency to address Challenge 3 (Sec. 3.2); and an *Endpoint Safety Projection* mechanism that ensures numerically valid endpoint updates to address Challenge 1 (Sec. 3.3).

## 3. Proposed Framework

We propose **SimGFM**, a minimalist framework for graph DFM that adheres strictly to the standard formulation without introducing auxiliary modules, thereby preserving fidelity to flow-matching theory. The overall pipeline is illustrated in Figure 2.

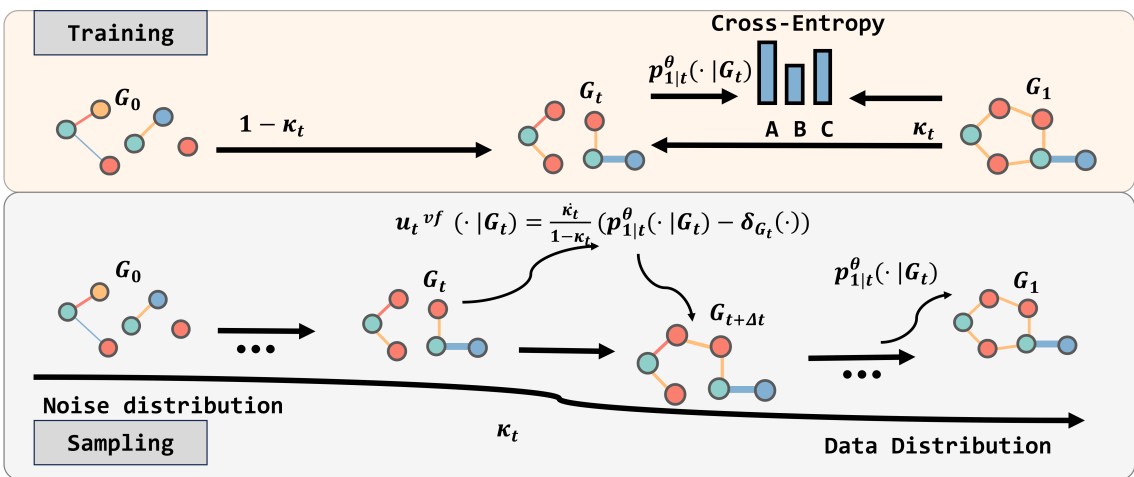

Figure 2. Our Proposed SimGFM Framework.

## 3.1. Velocity Field of SimGFM

In the DFM framework, the choice of the velocity field $u_t$ is central. Campbell's construction (Eq. 5), while theoretically sound, requires conditioning on fixed endpoints and averaging over posterior distributions, which incurs substantial computational overhead and hinders low-step generation. To alleviate this, we adopt the vf denoiser (Eq. 6) as our backbone, valued for its simplicity and scheduler-based flexibility.

**Implementation-friendly rate instantiation.** Campbell-style velocity fields either rely on expensive posterior-expectation computations or resort to Monte Carlo approximations: the former incurs high computational overhead, while the latter requires sufficiently many samples to control the estimator variance. In contrast, under the vf-denoiser framework, we instantiate the graph rate matrix using only $p_{1|t}^{\theta}(\cdot \mid G_t)$ and the current state, thereby avoiding explicit posterior expectations (and their Monte Carlo approximations). This design is straightforward to implement and more computationally efficient, and it reduces the propagation of sampling-induced errors in the low-step regime.

**Few-step sensitivity and stabilization.** As discussed by Boget (2025), DFM trajectories can exhibit *compounding denoising errors*, where small prediction errors accumulate and propagate along iterative denoising; this issue is more visible when the number of sampling steps is small. To mitigate this sensitivity in graph generation, SimGFM combines the above rate instantiation with an endpoint-focused scheduler that allocates denser updates as $t \to 1$ (Section 3.2).

## 3.2. The Choice of Scheduler

The temporal dynamics of discrete graph generation differ significantly from continuous domains. As illustrated in

Figure 3, we analyze the denoising trajectory on the Planar dataset and observe a critical phenomenon: under uniform denoising, valid graph structures emerge almost exclusively near the endpoint regime $t \to 1$. In the early and mid stages, structural validity remains close to zero, suggesting that a large portion of the computation budget is spent in regions that contribute little to the formation of valid structures. For completeness, we provide the validity–$t$ trajectories on Tree, QM9, and QM9H in Appendix Figure 6.

This empirical pattern motivates the use of a non-uniform scheduler that allocates more updates near $t = 1$, where graph validity is most sensitive. Building upon the time-distortion strategy popularized by DeFoG (QIN et al., 2025), we adopt a polynomial scheduler of the form $f_k(t) = 1 - (1-t)^k$ with $k \geq 1$ (Figure 4). A larger $k$ slows down noise progression in the endpoint region, allowing the model to devote finer-grained updates precisely where valid structures are formed. As shown in Figure 3, this leads to notably earlier and smoother emergence of valid graphs, in contrast to the sharp late-stage jump exhibited by baseline methods.

Importantly, while this scheduler substantially enhances the performance of our velocity-field-based formulation (Sec. 3.1), it is incompatible with Campbell's construction. Campbell's formulation was not derived with any scheduler in mind, and applying it with non-uniform schedules typically requires a time-distortion approximation. As analyzed in Appendix C, this approximation causes Campbell's updates to diminish rapidly under high-$k$ schedulers, leading to vanishing refinements in the endpoint region. In contrast, vf-denoiser maintains stable update magnitudes even under very high-order scheduling, enabling efficient targeted refinement and supporting high validity with significantly fewer denoising steps.

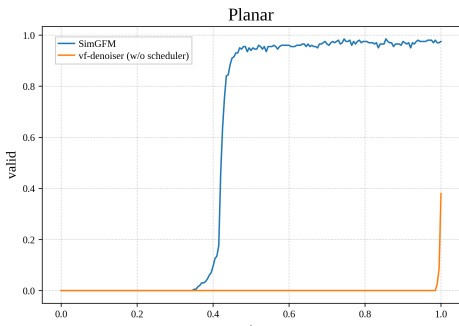

*Figure 3.* Vf-denoiser rises sharply at the very end, suggesting that valid graphs emerge late in denoising. SimGFM improves validity by allocating more steps to this critical late-stage refinement phase.

## 3.3. Endpoint Safety Projection

**Endpoint discretization requires correction.** We employ the polynomial scheduler $\kappa_t = 1 - (1 - t)^k$ (Sec. 3.2). As $t \to 1$, the factor $\dot{\kappa}_t/(1 - \kappa_t)$ is amplified, which can make the Euler discretization of the CTMC update numerically invalid (e.g., negative probability mass). The following theorem characterizes the set of steps that require an endpoint correction under a uniform time grid.

**Theorem 3.1** (Endpoint correction window under the polynomial scheduler). *Consider the vf-denoiser update in Eq. (6) and its Euler one-step kernel $q_t^i(x^i \mid z) = \delta_{z^i}(x^i) + \Delta t\, u_t^i(x^i, z)$. Let $\kappa_t = 1 - (1 - t)^k$ and use a uniform grid $t_n = n\Delta t$ with $\Delta t = 1/\texttt{maxstep}$. Then, for any coordinate $i$, the "stay" probability admits the lower bound*

$$q_{t_n}^i(z^i \mid z) \geq 1 - \Delta t \cdot \frac{\dot{\kappa}_{t_n}}{1 - \kappa_{t_n}} = 1 - \Delta t \cdot \frac{k}{1 - t_n}. \quad (11)$$

*Consequently, when $n > \texttt{maxstep} - k$ (i.e., the last $k$ steps), the Euler kernel no longer guarantees nonnegativity. A short proof is provided in Appendix A.*

**Endpoint Safety Projection (ESP).** Motivated by Theorem 3.1, we define an endpoint safeguard that is triggered in the tail window. Let

$$t_{\text{ESP}} := 1 - k\Delta t, \quad (12)$$

and for all $t \geq t_{\text{ESP}}$, we replace the CTMC Euler update (Eq. (9)) by directly sampling from the model posterior:

$$G_{t+\Delta t} \sim p_{1|t}^\theta(\cdot \mid G_t), \qquad \text{for } t \geq t_{\text{ESP}}. \quad (13)$$

Intuitively, in this endpoint window, the next-step marginal induced by the DFM Euler update $X_{t+\Delta t} \sim \delta_{X_t}(\cdot) + \Delta t\, u_t(\cdot, X_t)$ in Eq. (2) satisfies $p_{t+\Delta t}(\cdot) \to p_1(\cdot)$ as $t \to 1$. Meanwhile, the learned posterior $p_{1|t}^\theta(\cdot \mid X_t)$ in Eq. (7) is trained to approximate $p_{1|t}(\cdot \mid X_t)$ and also concentrates toward $p_1(\cdot)$ in the same limit. Therefore, when

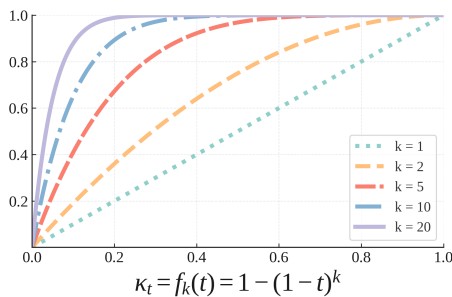

*Figure 4.* Polynomial scheduler curves. The polynomial scheduler $f_k(t)$ flattens near $t = 1$ at higher $k$, concentrating steps in the critical refinement region.

---

**Algorithm 1** SimGFM Training

---

1: **Input:** Graph dataset $\mathcal{D} = \{G^1, \ldots, G^M\}$
2: **while** $f_\theta$ not converged **do**
3:     Sample $G_1 \sim \mathcal{D}$
4:     Sample $t \sim \mathcal{T}$
5:     Sample $G_0 \sim p_0(G_0)$
6:     Sample $G_t \sim p_t(G_t \mid G_0, G_1)$
7:     $p_{1|t}^\theta(\cdot \mid G_t) \leftarrow f_\theta(G_t, t)$
8:     loss $\leftarrow \text{CE}_\lambda(G_1, p_{1|t}^\theta(\cdot \mid G_t))$
9:     optimizer.step(loss)
10: **end while**

---

$\kappa_t \to 1$, replacing the remaining DFM updates by directly sampling from $p_{1|t}^\theta(\cdot \mid X_t)$ is a reasonable endpoint approximation. Our polynomial scheduler satisfies this condition precisely: when the endpoint correction is triggered, $\kappa_t$ is already sufficiently close to 1. For instance, with $k = 10$ and $\texttt{maxstep} = 50$ (implying $\Delta t = 0.02$), the ESP window begins at $t_{\text{ESP}} = 0.80$. At this stage, $\kappa_{0.8} = 1 - (0.2)^{10} \approx 1 - 1.0 \times 10^{-7}$. From the DFM evolution trend, ESP is reasonable: in the late sampling stage, the endpoint posterior should gradually dominate the update; see Appendix A.2 for analysis. Meanwhile, Appendix A.3 proves from a scheduler-correction perspective that ESP is fully compatible with discrete flow matching theory.

## 3.4. Training and Sampling Procedures of SimGFM

Our framework follows the standard procedure of DFM (see Figure 2), but its core driving mechanism—the construction of the rate matrix—is redesigned to be more direct and efficient.

**Training.** We design the training procedure of **SimGFM** as shown in Algorithm 1. The entire objective centers on a single task: to teach a graph neural network $f_\theta$ to accurately predict the final, clean target graph $G_1$ from a halfway-evolved, ambiguous intermediate graph $G_t$. Each training

**Algorithm 2** SimGFM Sampling with ESP

1: **Input:** # graphs to sample $S$
2: Set $t_{\text{ESP}} \leftarrow 1 - k\Delta t$
3: **for** $i = 1$ to $S$ **do**
4:   Sample $N$ from train set
5:   Sample $G_0 \sim p_0(G_0)$
6:   **for** $t = 0$ to $1 - \Delta t$ **do**
7:     $p_{1|t}^{\theta}(\cdot \mid G_t) \leftarrow f_\theta(G_t, t)$
8:     **if** $t \geq t_{\text{ESP}}$ **then**
9:       $G_{t+\Delta t} \sim p_{1|t}^{\theta}(\cdot \mid G_t)$
10:     **else**
11:       $u_t(\cdot, G_t) \leftarrow \frac{\dot{\kappa}_t}{1-\kappa_t}\left[ p_{1|t}^{\theta}(\cdot \mid G_t) - \delta_{G_t}(\cdot) \right]$
12:       $G_{t+\Delta t} \sim \delta_{G_t}(\cdot) + \Delta t \cdot u_t(\cdot, G_t)$
13:     **end if**
14:   **end for**
15:   Store $G_1$
16: **end for**

iteration begins by sampling a real graph $G_1$ from the dataset and a time point $t$. An intermediate state $G_t$ between pure noise and real data is then generated according to **Eq. 8**. Next, the noised graph $G_t$, together with the current time $t$, is fed into the network to produce a prediction of the posterior distribution over the original graph $G_1$. Finally, we optimize the model parameters by computing the cross-entropy between this predicted distribution and the ground-truth graph.

$$
\mathcal{L}(\theta) = -\sum_{i\in[N]} \mathbb{E}_{t,(G_0,G_1),G_t} \log p_{1|t}^{\theta}\big(x_1^i \mid G_t\big) \; - \\
\sum_{1\leq i < j \leq N} \mathbb{E}_{t,(G_0,G_1),G_t} \log p_{1|t}^{\theta}\big(e_1^{ij} \mid G_t\big)
\tag{14}
$$

Here, $x_1^i$ denotes the attribute of the $i$-th node in $G_1$ (i.e., the target label), and $e_1^{ij}$ denotes the attribute of the node pair $(i, j)$ in $G_1$: the value 1 indicates that the edge is absent, while any other value represents the attribute of an existing edge.

**Sampling.** The sampling procedure of SimGFM (Algorithm 2) generates graphs by evolving a noise graph $G_0 \sim p_0$ from $t = 0$ to $t = 1$ with step size $\Delta t$. At each time $t$, the denoiser predicts the posterior distribution $p_{1|t}^{\theta}(\cdot \mid G_t)$. When $t < t_{\text{ESP}}$, we construct the executable velocity field via the vf-denoiser (Eq. (6)),

$$
u_t(\cdot, G_t) \leftarrow \frac{\dot{\kappa}_t}{1-\kappa_t}\left[ p_{1|t}^{\theta}(\cdot \mid G_t) - \delta_{G_t}(\cdot) \right],
$$

and perform a CTMC Euler update as in Eq. (9) to obtain $G_{t+\Delta t}$.

In the endpoint window $t \geq t_{\text{ESP}} := 1 - k\Delta t$, we activate *Endpoint Safety Projection* (ESP, Eq. (13)) and replace the CTMC update by directly sampling $G_{t+\Delta t} \sim p_{1|t}^{\theta}(\cdot \mid G_t)$. This safeguard is motivated by Theorem 3.1. Under the polynomial scheduler, $\kappa_t$ rapidly approaches 1, and the last

$k$ steps may suffer from negative-probability issues due to Euler discretization. We therefore treat this tail region as an endpoint regime and apply ESP as a numerical safeguard; see Sec. 3.3 for details.

## 4. Experiments

### 4.1. Experimental Setup

**Datasets.** We evaluate SimGFM across three task groups: (1) *generic graph generation* — Planar, SBM (Martinkus et al., 2022), Tree (Bergmeister et al., 2023), Ego-small, Community-small, Grid (Jo et al., 2022); (2) *molecular graph generation* — QM9 / QM9-with-H (Wu et al., 2018), MOSES (Polykovskiy et al., 2020); and (3) *conditional generation* — TLS (Madeira et al., 2024). Following prior work, we adopt the standard evaluation protocol for each dataset, reporting Valid/Unique/Novel (V.U.N.), Ratio, Fréchet ChemNet Distance (FCD), and graph statistics distances (Degree-MMD, Clustering-MMD, Orbit-MMD).

**Baselines.** We compare against major families of graph generative models. **Autoregressive models** include GraphRNN (You et al., 2018), GRAN (Liao et al., 2019), GraphGen (Goyal et al., 2020) BiGG (Dai et al., 2020), and AUTOGRAPH (Chen et al., 2025). **GAN models** cover GraphNVP (Madhawa et al., 2019) and SPEC-TRE (Martinkus et al., 2022). **Diffusion models** consist of DiGress (Vignac et al., 2022), GDSS (Jo et al., 2022), EDGE (Chen et al., 2023), BwR (Diamant et al., 2023), HSpectre (Bergmeister et al., 2023), GruM (Jo et al., 2023), DisCo (Xu et al., 2024), Cometh (Siraudin et al., 2024) and SID/CID (Boget, 2025) Finally, **Flow models** include De-FoG (QIN et al., 2025), CatFlow (Eijkelboom et al., 2024), and GGFlow (Hou et al., 2025).

Baseline results are from official implementations or reported numbers in the corresponding papers; further details in Appendix E.

### 4.2. Overall Performance

> Requiring only **10–50 sampling steps**, SimGFM can match or even outperform SOTA models across generic, molecular, and conditional graph generation tasks.

#### 4.2.1. GENERIC GRAPH GENERATION

We evaluate SimGFM on the standard Planar, SBM, and Tree benchmarks. Table 1 reports two key metrics: (i) valid/unique/novel (V.U.N.) graphs and (ii) the Ratio of graph-statistic distances between generated and test sets relative to the train–test distance (lower is better). SimGFM demonstrates strong efficiency: on **Planar**, it achieves **99.5%** V.U.N. with a Ratio of **1.8** using only **50** steps;

*Table 1.* Graph generation performance on the synthetic datasets: Planar, Tree and SBM. V.U.N. denotes Valid, Unique, and Novel, with Ratio closer to 1 indicating better alignment. Values are mean $\pm$ std from five runs of 40 graphs each. Best and second-best results are in bold and underline.

| Model | Class | # Steps ↓ | Planar | | Tree | | SBM | |
|---|---|---|---|---|---|---|---|---|
| | | | V.U.N. ↑ | Ratio ↓ | V.U.N. ↑ | Ratio ↓ | V.U.N. ↑ | Ratio ↓ |
| Train set | — | — | 100 | 1.0 | 100 | 1.0 | 85.9 | 1.0 |
| GraphRNN | Autoregressive | — | 0.0 | 490.2 | 0.0 | 607.0 | 5.0 | 14.7 |
| GRAN | Autoregressive | — | 0.0 | 2.0 | 0.0 | 607.0 | 25.0 | 9.7 |
| BiGG | Autoregressive | — | 5.0 | 16.0 | 75.0 | 5.2 | 10.0 | 11.9 |
| GraphGen | Autoregressive | — | 7.5 | 210.3 | 95.0 | 33.2 | 5.0 | 48.8 |
| AUTOGRAPH | Autoregressive | — | 87.5 | **1.5** | — | — | 92.5 | 3.4 |
| EDGE | Diffusion | 1000 | 0.0 | 431.4 | 0.0 | 850.7 | 0.0 | 51.4 |
| BwR (EDP-GNN) | Diffusion | 1000 | 0.0 | 251.9 | 0.0 | 11.4 | 7.5 | 38.6 |
| DiGress | Diffusion | 1000 | 77.5 | 5.1 | 90.0 | 1.6 | 60.0 | 1.7 |
| HSpectre | Diffusion | — | 95.0 | 2.1 | **100.0** | 4.0 | 75.0 | 10.5 |
| GruM | Diffusion | — | 90.0 | 1.8 | — | — | 85.0 | **1.1** |
| DisCo | Diffusion | 500 | 83.6 | — | — | — | 66.2 | — |
| Cometh | Diffusion | 500 | 92.5 | — | — | — | 77.0 | — |
| Cometh-PC | Diffusion | — | 99.5 | — | — | — | — | — |
| CatFlow | Flow | — | 80.0 | — | — | — | 85.0 | — |
| DeFoG (50 steps) | Flow | 50 | 95.0 | 3.2 | 73.5 | 2.5 | 86.5 | 2.2 |
| DeFoG (1000 steps) | Flow | 1000 | 99.5 | 1.6 | 96.5 | 1.6 | 90.0 | 4.9 |
| SimGFM (20 steps) | Flow | **20** | 94.0±4.4 | 2.3±0.6 | 88.0±4.8 | 2.5±0.9 | 82.0±4.0 | 5.6±1.1 |
| SimGFM (50 steps) | Flow | 50 | 99.5±1.0 | 1.8±0.5 | 97.0±1.0 | 2.0±0.7 | 87.0±4.0 | 2.9±0.5 |
| SimGFM (200 steps) | Flow | 200 | **100.0**±0.0 | 9.3±2.6 | 99.5±1.0 | **1.5**±0.2 | 90.5±4.0 | 3.2±0.5 |
| SimGFM (1000 steps) | Flow | 1000 | **100.0**±0.0 | 21.0±1.5 | **100.0**±0.0 | 2.1±0.8 | **93.0**±1.9 | 8.2±3.0 |

on **Tree**, it reaches **99.5**% V.U.N. and **1.5** Ratio at **200** steps; and on **SBM**, it matches the performance of DeFoG with **200** steps, compared to DeFoG's **1000**. For a matched-step comparison against the 1000-step baselines, SimGFM reaches $100.0\%$ V.U.N. on Planar and Tree, and $93.0\%$ on SBM, with Ratios 21.0, 2.1, and 8.2, respectively. The unusually high-validity but weaker-Ratio behavior on Planar at large step counts is further analyzed in Appendix F.7 and Appendix F.8. These results highlight that a minimalist, well-founded design can deliver both competitiveness and efficiency.

We further assess structural fidelity on Ego-small, Community-small, and Grid (see Appendix F.3).We also test conditional generation on TLS (see Appendix F.5).

### 4.2.2. MOLECULAR GRAPH GENERATION

We further evaluate SimGFM on three molecular benchmarks. On **QM9**, Table 2 shows that SimGFM achieves SOTA performance at **200** steps, while already reaching **99.5**% validity with only **10** steps, which is an order of magnitude fewer than the $\sim500$ steps typically required by diffusion models, thereby demonstrating substantial gains in sampling efficiency. On **QM9-with-H**, results in Table 2 indicate that SimGFM at **200** steps matches or surpasses the best reported scores across all metrics, and at just **50** steps achieves a FCD of **0.04**. Under a matched budget of 500 steps, SimGFM attains $99.8\%$ validity with FCD 0.31 on

QM9 and $98.6\%$ validity with FCD 0.04 on QM9-with-H, confirming that the advantage is not limited to the aggressive low-step regime.

For the large-molecule benchmarks **MOSES** and **Guacamol**, Table 3 shows that SimGFM remains highly effective in the low-step regime. On **Guacamol**, SimGFM already delivers strong results at just **50** steps (Val.=98.9, V.U.N.=98.3) and attains the **best FCD** among all compared methods, while further improving to 99.2 validity at **200** steps, demonstrating high-fidelity large-molecule generation with substantially fewer denoising iterations. On **MOSES**, SimGFM at **200** steps improves validity from 86.1 to 89.4 and achieves a lower FCD of **1.08** while keeping **100.0** uniqueness, indicating stronger distribution matching without sacrificing diversity. Even in the more extreme low-step regime, SimGFM remains competitive: at 10/20 steps it achieves 85.2/94.5 validity with FCD 55.3/73.2 on GuacaMol, and 79.3/84.0 validity with FCD 1.14/1.11 on MOSES. At the matched budget of 500 steps, SimGFM further reaches $98.6\%$ V.U.N. with FCD 79.7 on Guacamol and $91.1\%$ validity with FCD 1.19 on MOSES.

### 4.3. Sampling Efficency

We report validity and FCD as functions of sampling steps on QM9 (Figs. 8a and 8b in appendix). SimGFM surpasses 0.99 validity with only **10** steps, whereas other methods typically require at least **50**. This advantage arises from

*Table 2.* Molecule generation on QM9. We present the results over five sampling runs of 10000 generated graphs each. We include the results of Relaxed Validity, which accounts for charged molecules, to facilitate comparison.

| | Without Explicit Hydrogens | | | | | With Explicit Hydrogens | | | | |
|---|---|---|---|---|---|---|---|---|---|---|
| **Model** | # Steps ↓ | Valid ↑ | Relaxed Valid ↑ | Unique ↑ | FCD ↓ | # Steps ↓ | Valid ↑ | Relaxed Valid ↑ | Unique ↑ | FCD ↓ |
| Training set | — | 99.3 | 99.5 | 99.2 | 0.03 | — | 97.8 | 98.9 | 99.9 | 0.01 |
| SPECTRE | — | 87.3 | — | 35.7 | — | — | — | — | — | — |
| GraphNVP | — | 83.1 | — | 99.2 | — | — | — | — | — | — |
| GDSS | — | 95.7 | — | 98.5 | 2.9 | — | — | — | — | — |
| DiGress | — | 99.0 | — | 96.2 | — | — | 95.4 | — | **97.6** | — |
| GruM | — | 99.2 | — | 96.7 | **0.11** | — | — | — | — | — |
| CatFlow | — | 99.8 | — | **100.0** | 0.44 | — | — | — | — | — |
| DisCo | — | 99.3 | — | — | — | — | — | — | — | — |
| Cometh | — | 99.6 | — | 96.8 | 0.25 | — | — | — | — | — |
| GRAPHARM | — | 90.25 | — | 95.62 | 1.22 | — | — | — | — | — |
| SID | — | 99.7 | — | — | 0.50 | — | — | — | — | — |
| CID | — | **99.9** | — | — | 1.76 | — | — | — | — | — |
| DeFoG (50 steps) | 50 | 98.9 | 99.2 | 96.2 | 0.26 | 50 | 97.1 | 98.1 | 94.8 | 0.31 |
| DeFoG (500 steps) | 500 | 99.3 | 99.4 | 96.3 | 0.12 | 500 | 98.0 | 98.8 | 96.7 | 0.05 |
| SimGFM (10 steps) | **10** | 99.5±0.0 | 99.7±0.0 | 95.0±0.2 | 0.92±0.0 | **10** | 93.7±0.2 | 95.6±0.3 | **97.6**±0.1 | 0.07±0.0 |
| SimGFM (50 steps) | 50 | 99.7±0.0 | 99.8±0.0 | 96.3±0.0 | 0.13±0.0 | 50 | 98.4±0.0 | 99.2±0.1 | 97.1±0.1 | **0.04**±0.0 |
| SimGFM (200 steps) | 200 | 99.8±0.0 | 99.8±0.0 | 95.9±0.0 | 0.15±0.0 | 200 | 98.4±0.1 | 99.2±0.0 | 97.0±0.3 | **0.04**±0.0 |
| SimGFM (500 steps) | 500 | 99.8±0.0 | **99.9**±0.0 | 94.9±0.0 | 0.31±0.0 | 500 | **98.6**±0.0 | **99.3**±0.0 | 96.9±0.0 | **0.04**±0.0 |

*Table 3.* Large molecule generation performance. Only iterative denoising-based methods are reported here.

| | Guacamol | | | | | MOSES | | | | | | |
|---|---|---|---|---|---|---|---|---|---|---|---|---|
| **Model** | Val. ↑ | V.U. ↑ | V.U.N. ↑ | KL div ↑ | FCD ↑ | Val. ↑ | Unique. ↑ | Novelty ↑ | Filters ↑ | FCD ↓ | SNN ↑ | Scaf ↑ |
| Training set | 100.0 | 100.0 | 0.0 | 99.9 | 92.8 | 100.0 | 100.0 | 0.0 | 100.0 | 0.01 | 0.64 | 99.1 |
| DiGress | 85.2 | 85.2 | 85.1 | 92.9 | 68.0 | 85.7 | **100.0** | 95.0 | 97.1 | 1.19 | 0.52 | 14.8 |
| DisCo | 86.6 | 86.6 | 86.5 | 92.6 | 59.7 | 88.3 | **100.0** | 97.7 | 95.6 | 1.44 | 0.50 | 15.1 |
| Cometh | 98.9 | 98.9 | 97.6 | 96.7 | 72.7 | 90.5 | 99.9 | 92.6 | 99.1 | 1.27 | 0.54 | 16.0 |
| DeFoG (50 steps) | 91.7 | 91.7 | 91.2 | 92.3 | 57.9 | 83.9 | 99.9 | 96.9 | 96.5 | 1.87 | 0.50 | **23.5** |
| DeFoG (500 steps) | 99.0 | 99.0 | 97.9 | **97.7** | 73.8 | **92.8** | 99.9 | 92.1 | 98.9 | 1.95 | 0.55 | 14.4 |
| SimGFM (10 steps) | 85.2 | 85.2 | 85.0 | 89.5 | 55.3 | 79.3 | **100.0** | **97.8** | 95.6 | 1.14 | 0.49 | 22.0 |
| SimGFM (20 steps) | 94.5 | 94.5 | 94.2 | 91.7 | 73.2 | 84.0 | **100.0** | 96.0 | 98.0 | 1.11 | 0.53 | 19.4 |
| SimGFM (50 steps) | 98.9 | 98.9 | 98.3 | 93.6 | 79.6 | 86.1 | **100.0** | 97.5 | 95.3 | 1.44 | 0.51 | 16.9 |
| SimGFM (200 steps) | **99.2** | **99.2** | **98.8** | 94.0 | **80.8** | 89.4 | **100.0** | 94.6 | 98.9 | **1.08** | 0.55 | 17.3 |
| SimGFM (500 steps) | **99.2** | **99.2** | 98.6 | 94.6 | 79.7 | 91.1 | **100.0** | 93.5 | 99.2 | 1.19 | **0.56** | 14.4 |

the DFM mechanism: by following a straighter probability path, SimGFM reaches high validity with substantially fewer refinement steps.

In terms of FCD (Fig. 8b), SimGFM decays rapidly from **0.92** at 10 to **0.15** at 200. Thus, **10** steps already attain performance once associated with **500 − 1000**, while **200** steps achieve best-in-class results, underscoring a significant improvement in sampling efficiency.

### 4.4. Ablation Study

We conduct ablations on two sampling sensitive components: (i) the Endpoint Safety Projection (ESP; Sec. 3.3), which stabilizes the Euler mixture update in the endpoint region as $t \to 1$; and (ii) the time scheduler $\kappa_t$.

Appendix F.6 summarizes the results. Table 10 separately evaluates the effect of enabling ESP on several benchmarks; Table 11 further studies the interaction between ESP and the scheduler: under the same vf-denoiser backbone, we compare the identity scheduler ($k = 1$) with our proposed scheduler.

Even with the identity scheduler ($k = 1$), its endpoint focus is clearly weaker than that of a high-order polynomial scheduler (large $k$), but the last step of the discrete Euler update still enters the endpoint region, making the "stay" coefficient $(1 - \lambda_t)$ in the mixture become extremely small and potentially cross zero, thereby causing negative probabilities in the transition kernel. Therefore, under the setting $k = 1$, ESP can still stably improve performance, and it typically yields larger gains when combined with an endpoint-focused scheduler.

## 5. Related Work

**Diffusion models** (Ho et al., 2020) treat generation as iterative denoising. Discrete variants like DiGress (Vignac et al., 2022) edit nodes and edges categorically while preserving marginals, achieving strong results on molecular and non-molecular datasets. Extensions such as EDGE (Chen et al., 2023), and DisCo (Xu et al., 2024) improve efficiency or structural modeling through mixture strategies, bandwidth constraints, or richer encodings. SID (Boget, 2025) partially mitigates compounding denoising errors by assum-

ing conditional independence between intermediate states. Continuous-time variants (Campbell et al., 2022; Xu et al., 2024) employ CTMCs; e.g., Cometh (Siraudin et al., 2024) integrates random-walk features to boost validity, uniqueness, and novelty. Despite these advances, diffusion remains hindered by slow sampling and broader error accumulation.

**Flow Matching (FM)** offers a more efficient refinement paradigm, transporting noise to data via ODEs or CTMCs with improved stability (Lipman et al., 2022; Liu et al., 2022) and demonstrated success in vision domains (Esser et al., 2024; Ma et al., 2024). Its discrete extension, **DFM** (Campbell et al., 2024; Gat et al., 2024), extends the framework to categorical data, including graphs, by employing linear interpolation and CTMC dynamics. Subsequent works such as CatFlow (Hou et al., 2025), DeFoG (QIN et al., 2025), and GGFlow (Hou et al., 2025) enhance performance but rely on costly optimization, heuristics, or reinforcement learning, complicating the framework.

## 6. Conclusion

We present SimGFM, a minimal yet strong framework for discrete flow matching on graphs, built on a clean CTMC formulation, a monotone scheduler, and a compatible Endpoint Safety Projection. With only these design choices, SimGFM can match or surpass more complex methods with just 10–50 sampling steps, substantially improving sampling efficiency while maintaining strong performance.

## Acknowledgments

This work received support the "111 Center" (No. B26023), Fundamental and Interdisciplinary Disciplines Breakthrough Plan of the Ministry of Education of China (No. JYB2025XDXM118), NSFC Grant 62572026, National Social Science Fund of China 22&ZD153, and State Key Laboratory of Complex & Critical Software Environment (SKLCCSE). This work was primarily conducted at Nanjing University, which serves as the lead affiliation. Yuankai Luo and Lei Shi are the corresponding authors.

## Impact Statement

This paper presents work whose goal is to advance the field of Graph Machine Learning. There are many potential societal consequences of our work, none of which we feel must be specifically highlighted here.

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

# A. Analysis of ESP

## A.1. Proof of Theorem 3.1

We rewrite the Euler one-step update induced by the vf-denoiser into a convex-combination form. This makes explicit when the discretized kernel may assign negative mass, motivating ESP.

**Euler one-step kernel as a convex combination.**   Consider the Euler CTMC discretization in Eq. (2) with vf-denoiser (Eq. (6)):

$$q_t(\cdot \mid X_t) := \delta_{X_t}(\cdot) + \Delta t\, u_t(\cdot, X_t), \qquad u_t(\cdot, X_t) = \frac{\dot{\kappa}_t}{1 - \kappa_t}\left[p_{1|t}^{\theta}(\cdot \mid X_t) - \delta_{X_t}(\cdot)\right].$$

Substituting $u_t$ gives

$$q_t(\cdot \mid X_t) = \left(1 - \lambda_t\right)\delta_{X_t}(\cdot) + \lambda_t\, p_{1|t}^{\theta}(\cdot \mid X_t), \qquad \lambda_t := \Delta t \cdot \frac{\dot{\kappa}_t}{1 - \kappa_t}. \tag{1}$$

Hence the induced next-step marginal satisfies

$$p_{t+\Delta t}(x) = \mathbb{E}_{X_t \sim p_t}\left[q_t(x \mid X_t)\right] = \mathbb{E}_{X_t \sim p_t}\left[(1 - \lambda_t)\delta_{X_t}(x) + \lambda_t\, p_{1|t}^{\theta}(x \mid X_t)\right]. \tag{2}$$

**Nonnegativity requires $1 - \lambda_t \geq 0$.**   Since $\delta_{X_t}$ and $p_{1|t}^{\theta}(\cdot \mid X_t)$ are nonnegative measures, the one-step kernel in Eq. (1) is a valid probability kernel only if its coefficients are nonnegative, i.e.,

$$1 - \lambda_t \geq 0 \quad \Longleftrightarrow \quad \lambda_t \leq 1 \quad \Longleftrightarrow \quad \Delta t \cdot \frac{\dot{\kappa}_t}{1 - \kappa_t} \leq 1. \tag{3}$$

If $\lambda_t > 1$, then the coefficient $(1 - \lambda_t)$ becomes negative and the discretized kernel may assign negative probability mass (numerically observed as "negative probabilities").

**Polynomial scheduler implies a last-$k$-steps risk window.**   With the polynomial scheduler $\kappa_t = 1 - (1 - t)^k$,

$$\frac{\dot{\kappa}_t}{1 - \kappa_t} = \frac{k(1 - t)^{k-1}}{(1 - t)^k} = \frac{k}{1 - t}. \tag{4}$$

Using a uniform grid $t_n = n\Delta t$ with $\Delta t = 1/\texttt{maxstep}$, we have

$$\lambda_{t_n} = \Delta t \cdot \frac{\dot{\kappa}_{t_n}}{1 - \kappa_{t_n}} = \Delta t \cdot \frac{k}{1 - t_n}. \tag{5}$$

The validity condition in Eq. (3) is violated whenever

$$\lambda_{t_n} > 1 \quad \Longleftrightarrow \quad \Delta t \cdot \frac{k}{1 - t_n} > 1 \quad \Longleftrightarrow \quad 1 - t_n < k\Delta t \quad \Longleftrightarrow \quad n > \texttt{maxstep} - k,$$

which corresponds exactly to the last $k$ steps. Therefore, under the polynomial scheduler, Euler discretization may fail to guarantee nonnegativity in the endpoint tail window, motivating an endpoint safeguard such as ESP (Eq. (13)). □

## A.2. Why the posterior-transfer coefficient should dominate near the endpoint

The coefficient $\lambda_t$ in Eq. (1) has a direct interpretation: it is the amount of mass transferred from the current state $\delta_{X_t}$ to the predicted endpoint posterior $p_{1|t}^{\theta}(\cdot \mid X_t)$ in one Euler step. Thus, increasing $\lambda_t$ means relying less on staying at the noisy state and more on the model's endpoint prediction. This is the intended behavior near $t = 1$, where the generation process should perform final structural correction rather than preserve noise.

For the polynomial scheduler $\kappa_t = 1 - (1 - t)^k$, Eq. (5) gives

$$\lambda_{t_n} = \Delta t \frac{k}{1 - t_n}. \tag{6}$$

Hence $\lambda_t$ grows monotonically as $t$ approaches 1. This growth is not an accident; it encodes the inductive bias that endpoint information should dominate late in sampling. The only problem is that a finite Euler step can overshoot the probability boundary when $\lambda_t > 1$. ESP keeps the same bias but prevents the overshoot: it allows the endpoint posterior to receive full weight, while never allowing an invalid weight larger than one.

## A.3. A Scheduler-Correction View of ESP

**ESP as a corrected scheduler on the discrete grid.** The same calculation also yields a direct scheduler-correction interpretation of ESP. Define the corrected coefficient

$$\lambda'_t := \min(\lambda_t, 1). \tag{7}$$

Since $\lambda_t = \Delta t\, \dot{\kappa}_t / (1 - \kappa_t)$, this is equivalent to replacing the derivative by

$$\dot{\kappa}'_t := \min\left(\dot{\kappa}_t, \frac{1 - \kappa_t}{\Delta t}\right), \tag{8}$$

while keeping the current scheduler value $\kappa'_t = \kappa_t$ on the grid point $t = t_n$.

**Proposition A.1** (ESP equals a validity-preserving scheduler correction). *Let $q'_t(\cdot \mid X_t)$ denote the Euler kernel obtained by replacing $\lambda_t$ in Eq. (1) with $\lambda'_t$ from Eq. (7). Then:*

$$q'_t(\cdot \mid X_t) = \left(1 - \lambda'_t\right)\delta_{X_t}(\cdot) + \lambda'_t p^\theta_{1|t}(\cdot \mid X_t), \tag{9}$$

*with $0 \leq \lambda'_t \leq 1$. Hence $q'_t(\cdot \mid X_t)$ is always a valid probability kernel; it coincides with the original Euler step when $\lambda_t \leq 1$, and reduces to direct posterior sampling when $\lambda_t > 1$.*

*Proof.* By definition, $0 \leq \lambda'_t \leq 1$, so the coefficients in Eq. (9) are nonnegative and sum to 1. Therefore $q'_t(\cdot \mid X_t)$ is a valid convex combination of $\delta_{X_t}$ and $p^\theta_{1|t}(\cdot \mid X_t)$. If $\lambda_t \leq 1$, then $\lambda'_t = \lambda_t$, so the corrected kernel equals the original Euler kernel in Eq. (1). If $\lambda_t > 1$, then $\lambda'_t = 1$, hence $q'_t(\cdot \mid X_t) = p^\theta_{1|t}(\cdot \mid X_t)$, which is exactly the ESP rule in Eq. (13). Finally, Eq. (8) follows directly from solving $\lambda'_t = \Delta t\, \dot{\kappa}'_t / (1 - \kappa_t)$ under the constraint $\lambda'_t \leq 1$. $\square$

This proposition formalizes ESP as an internal scheduler correction rather than an external heuristic patch: it keeps the monotone bias toward stronger endpoint updates, but clips the update coefficient exactly at the boundary where the Euler discretization would otherwise become invalid. If one prefers a continuous scheduler representation, a monotone interpolation can be built through the corrected grid values induced by $\dot{\kappa}'_t$.

## A.4. Why ESP has limited impact on diversity

ESP is applied only in the endpoint tail window identified by Theorem 3.1. Before this point, the sampling trajectory has already passed through many stochastic transitions, so most global graph-level diversity has been established before ESP is triggered. The role of ESP is therefore local endpoint correction: it removes invalid negative mass in the final Euler steps rather than replacing the whole generative trajectory.

Moreover, ESP samples from the full model posterior $p^\theta_{1|t}(\cdot \mid X_t)$. It is not deterministic argmax decoding and therefore does not collapse all states to a single mode. Empirically, this is consistent with the ESP ablation in Appendix F.6: FCD changes negligibly while validity improves substantially. Since FCD is sensitive to diversity loss, a severe diversity collapse would cause a much larger degradation than what we observe.

# B. Permutation Invariance Guarantees

## B.1. Proof of Permutation Invariance for vf-denoiser with ESP

### B.1.1. NOTATION AND SETUP

We denote an undirected graph by $G = (x_{1:N}, e_{1 \leq i < j \leq N})$, where node variables take values in $\mathcal{X}$ and edge variables in $\mathcal{E}$. For any node permutation with permutation matrix $P \in \{0, 1\}^{N \times N}$, define the relabeling map $\pi_P$ acting on graph-indexed tensors by

$$\pi_P(X) = PX, \qquad \pi_P(A) = PAP^\top, \qquad [\pi_P(E)]_{\{i,j\}} = E_{\{P^{-1}(i), P^{-1}(j)\}}.$$

Let $\delta_G$ denote the Dirac measure at $G$, and $(\pi_P)_\# \mu$ the pushforward of a measure $\mu$ through $\pi_P$. Scalars such as $t, \Delta t, \kappa_t, \dot{\kappa}_t$ are invariant under $\pi_P$. We write $G_t$ for a noisy state, $f_\theta$ for the denoiser, and $p^\theta_{1|t}(\cdot \mid G_t)$ for the predicted clean-graph distribution.

### B.1.2. BACKBONE EQUIVARIANCE

**Proposition B.1** (Equivariance of the denoiser). *The attention-based Graph Transformer denoiser is permutation-equivariant:*

$$f_\theta(\pi_P(G_t), t) = \pi_P\big(f_\theta(G_t, t)\big).$$

*Consequently, the predicted distribution is equivariant in the sense of pushforward:*

$$p_{1|t}^\theta(\cdot \mid \pi_P(G_t)) = (\pi_P)_\# p_{1|t}^\theta(\cdot \mid G_t).$$

*Proof.* With shared projections $Q = XW_Q$, $K = XW_K$, $V = XW_V$, relabeling yields $Q' = PQ$, $K' = PK$, $V' = PV$. Shared edge bias/mask obeys $B' = PBP^\top$, $M' = PMP^\top$. The score matrix satisfies $L' = \frac{Q'K'^\top}{\sqrt{d_k}} + B' + M' = PLP^\top$. Row-softmax commutes with row permutations, hence $\mathrm{Att}' = P\,\mathrm{Att}\,P^\top$. Aggregation gives $Y' = \mathrm{Att}'V' = P(\mathrm{Att}V) = PY$. Pointwise residuals, layer normalizations, and MLPs commute with $P$. Multi-head attention and stacking preserve equivariance. The distribution statement follows because the output logits/tensors for nodes/edges are reindexed by $\pi_P$, so sampling a labeled graph under the permuted input is equivalent to pushing forward samples under the original input. $\square$

### B.1.3. LOSS INVARIANCE

**Proposition B.2** (Permutation-invariant training loss). *The training loss*

$$\mathcal{L}(\theta; G_t, G_1) = -\sum_{i \in [N]} \log p_{1|t}^\theta(x_1^i \mid G_t) - \sum_{1 \le i < j \le N} \log p_{1|t}^\theta(e_1^{ij} \mid G_t)$$

*is permutation-invariant:*

$$\mathcal{L}(\theta; \pi_P(G_t), \pi_P(G_1)) = \mathcal{L}(\theta; G_t, G_1).$$

*Proof.* By Proposition B.1, the node/edge predictive distributions under $\pi_P(G_t)$ are exactly reindexings of those under $G_t$. The node sum reindexes via $i \mapsto P(i)$, and the unordered edge sum reindexes via $\{i, j\} \mapsto \{P(i), P(j)\}$. Reindexing does not change the total, proving invariance. $\square$

### B.1.4. ONE-STEP KERNEL EQUIVARIANCE: VF-DENOISER AND ESP

We define the vf-denoiser Euler kernel (Eq. (2)) in measure form:

$$u_t^{\mathrm{vf}}(\cdot, G_t) = \frac{\dot{\kappa}_t}{1 - \kappa_t}\Big[p_{1|t}^\theta(\cdot \mid G_t) - \delta_{G_t}(\cdot)\Big], \qquad K_t^{\mathrm{vf}}(G_t, \cdot) = \delta_{G_t}(\cdot) + \Delta t\, u_t^{\mathrm{vf}}(\cdot, G_t).$$

For the endpoint safeguard (ESP, Eq. (13)), we use the direct-sampling kernel

$$K_t^{\mathrm{ESP}}(G_t, \cdot) = p_{1|t}^\theta(\cdot \mid G_t), \qquad \text{activated when } t \ge t_{\mathrm{ESP}}.$$

**Proposition B.3** (Equivariance of the vf-denoiser kernel). *For any measurable set $\mathcal{S}$,*

$$K_t^{\mathrm{vf}}(\pi_P(G_t), \pi_P(\mathcal{S})) = K_t^{\mathrm{vf}}(G_t, \mathcal{S}).$$

*Proof.* By Proposition B.1, $p_{1|t}^\theta(\cdot \mid \pi_P(G_t)) = (\pi_P)_\# p_{1|t}^\theta(\cdot \mid G_t)$, and trivially $(\pi_P)_\# \delta_{G_t} = \delta_{\pi_P(G_t)}$. Since $\kappa_t, \dot{\kappa}_t, \Delta t$ are scalars invariant under $\pi_P$, we obtain

$$(\pi_P)_\# u_t^{\mathrm{vf}}(\cdot, G_t) = u_t^{\mathrm{vf}}(\cdot, \pi_P(G_t)), \qquad (\pi_P)_\# K_t^{\mathrm{vf}}(G_t, \cdot) = K_t^{\mathrm{vf}}(\pi_P(G_t), \cdot).$$

Evaluating both sides on $\pi_P(\mathcal{S})$ yields the claim. $\square$

**Proposition B.4** (Equivariance of the ESP kernel). *For any measurable set $\mathcal{S}$,*

$$K_t^{\mathrm{ESP}}(\pi_P(G_t), \pi_P(\mathcal{S})) = K_t^{\mathrm{ESP}}(G_t, \mathcal{S}).$$

*Proof.* Directly from Proposition B.1:

$$K_t^{\mathrm{ESP}}(\pi_P(G_t), \pi_P(\mathcal{S})) = p_{1|t}^\theta(\pi_P(\mathcal{S}) \mid \pi_P(G_t)) = (\pi_P)_\# p_{1|t}^\theta(\mathcal{S} \mid G_t) = p_{1|t}^\theta(\mathcal{S} \mid G_t) = K_t^{\mathrm{ESP}}(G_t, \mathcal{S}).$$

$\square$

B.1.5. SAMPLING TRAJECTORY AND TRAINING OBJECTIVE

**Sampling invariance.** Assume the prior $p_0$ and the noising kernel $p_t(G_t \mid G_0, G_1)$ are compatible with permutations (relabeling only changes indices). Consider the (possibly time-inhomogeneous) sampling process that uses $K_t^{\mathrm{vf}}$ when $t < t_{\mathrm{ESP}}$ and $K_t^{\mathrm{ESP}}$ when $t \geq t_{\mathrm{ESP}}$. By Propositions B.3–B.4, each one-step kernel is permutation-equivariant; therefore, by induction over time steps and the Markov property, for any time grid and measurable sets $\{\mathcal{S}_\ell\}$,

$$\Pr\big(G_{t_1} \in \mathcal{S}_1, \ldots, G_{t_m} \in \mathcal{S}_m\big) = \Pr\big(\pi_P(G_{t_1}) \in \pi_P(\mathcal{S}_1), \ldots, \pi_P(G_{t_m}) \in \pi_P(\mathcal{S}_m)\big),$$

so the terminal sampling distribution (over isomorphism classes) is permutation-invariant.

**Training invariance.** Proposition B.2 shows the per-sample loss is permutation-invariant. Taking expectation over $(t, (G_0, G_1), G_t)$ preserves invariance; hence the overall training objective and its expected gradient are unchanged under node relabeling.

# C. Theoretical Analysis of Scheduler Compatibility and Update Dynamics

In this section, we provide the theoretical motivation for our choice of scheduler and strictly analyze the numerical behavior of different discrete flow matching formulations near the terminal time $t \to 1$.

## C.1. Motivation: The Necessity of Non-Linear Schedulers

Empirical observations on discrete graph generation (as discussed in Method) reveal a critical dynamical property: valid graph structures typically emerge only when the diffusion time $t$ is very close to 1. Consequently, a linear scheduler often wastes computational budget on early noisy stages. To address this, we employ a polynomial scheduler of the form:

$$f_k(t) = 1 - (1 - t)^k, \quad k \geq 1. \tag{10}$$

Let $\kappa_t = f_k(t)$. A larger $k$ (e.g., $k = 10$ or $20$) flattens the trajectory near $t = 1$, effectively increasing the sampling resolution in the region where structural validity is determined.

## C.2. Incompatibility of Time-Distortion Approximations (Campbell's Formulation)

(Campbell et al., 2024) proposed a discrete flow matching update based on *time distortion*. We prove here that this approximation suffers from vanishing updates when combined with the necessary high-$k$ schedulers derived above.

The inference process under time distortion approximates the flow by adjusting the time step magnitude based on $\kappa_t$. The update rule implies a transition proportional to the change in noise level:

$$x_{\kappa_{t+h}} \sim x_{\kappa_t} + (\kappa_{t+h} - \kappa_t) \cdot R, \tag{11}$$

where $R$ represents the rate or update direction. To analyze the behavior as $h \to 0$, we perform a Taylor expansion of the scheduler $f_k(t)$ around $t$:

$$\kappa_{t+h} = f_k(t + h) = f_k(t) + f_k'(t)h + O(h^2). \tag{12}$$

Substituting the derivative $f_k'(t) = k(1 - t)^{k-1}$, the effective update magnitude becomes:

$$\Delta \kappa \approx \kappa_{t+h} - \kappa_t = k(1 - t)^{k-1}h. \tag{13}$$

**Analysis as $t \to 1$:** When utilizing a scheduler with a large $k$ to improve validity, the term $(1 - t)^{k-1}$ approaches zero extremely rapidly as $t \to 1$. Consequently, the update probability mass $\Delta \kappa$ vanishes. This causes the sampling trajectory to "freeze" prematurely—the model fails to execute necessary structural refinements in the final steps because the effective step size under time distortion becomes numerically negligible.

## C.3. Robustness of the Velocity-Field Formulation (SimGFM)

In contrast, our proposed method (SimGFM) directly models the velocity field. The solver update rule for vf-denoiser is governed by the ratio of the rate of change to the remaining noise budget:

$$x_{t+h} \sim \delta_{x_t}(\cdot) + h \frac{\dot{\kappa}_t}{1 - \kappa_t} u, \tag{14}$$

where $u$ is the conditional vector field. Substituting the definitions for the polynomial scheduler $\kappa_t = 1 - (1 - t)^k$:

- The numerator: $\dot{\kappa}_t = k(1 - t)^{k-1}$.

- The denominator: $1 - \kappa_t = (1 - t)^k$.

The update coefficient simplifies to:

$$\frac{\dot{\kappa}_t}{1 - \kappa_t} = \frac{k(1 - t)^{k-1}}{(1 - t)^k} = \frac{k}{1 - t}. \tag{15}$$

Thus, the effective update rule behaves as:

$$x_{t+h} \sim \delta_{x_t}(\cdot) + h \frac{k}{1 - t} u. \tag{16}$$

**Conclusion:** Unlike the time-distortion formulation, the coefficient $h\frac{k}{1-t}$ does not vanish as $t \to 1$; instead, it compensates for the shrinking time horizon. This ensures that even with large $k$ values, the model maintains a significant probability of updating the graph structure up until the very end of the generation process.

## D. Two-stage sampling as an implicit ESP

We show that rvf-style two-stage sampling, together with a standard clipping-and-renormalization safeguard, behaves as an implicit endpoint safety projection (ESP) in the endpoint regime.

For completeness, we recall the rvf-denoiser used in this appendix. It first samples a candidate endpoint $\hat{x}_{1|t}^i \sim p_{1|t}^\theta(\cdot \mid z)$, and then replaces the dense posterior average by the following sparse velocity field:

$$u_t^{\mathbf{rvf},i}(x^i, z) = \frac{\dot{\kappa}_t}{1 - \kappa_t} \left[ \delta_{\hat{x}_{1|t}^i}(x^i) - \delta_{z^i}(x^i) \right]. \tag{17}$$

**Dense vf kernel.** Plugging Eq. (6) into the Euler update Eq. (2) yields a *dense* one-step categorical kernel (for each site $i$). Define

$$\lambda_t := \Delta t \cdot \frac{\dot{\kappa}_t}{1 - \kappa_t}, \qquad q_t^{\mathrm{vf}}(x^i \mid z) = (1 - \lambda_t)\, \delta_{z^i}(x^i) + \lambda_t\, p_{1|t}^\theta(x^i \mid z). \tag{18}$$

When $1 - \lambda_t \geq 0$, $q_t^{\mathrm{vf}}(\cdot \mid z)$ is a valid probability vector. In the endpoint tail window, however, $\lambda_t$ can exceed 1 (cf. Theorem 3.1), making the "stay" mass $1 - \lambda_t$ negative.

**Sparse rvf kernel via two-stage sampling.** In contrast, rvf-denoiser performs *two-stage sampling*: it first draws a candidate endpoint $\hat{x}_{1|t}^i \sim p_{1|t}^\theta(\cdot \mid z)$, and then applies a *sparse* Euler update (Eq. (17)), which corresponds to the conditional kernel

$$q_t^{\mathrm{rvf}}(x^i \mid z, \hat{x}_{1|t}^i) = (1 - \lambda_t)\, \delta_{z^i}(x^i) + \lambda_t\, \delta_{\hat{x}_{1|t}^i}(x^i). \tag{19}$$

This kernel is supported on at most two states, hence it is straightforward to sample from and to safeguard numerically.

**Same transitional kernel(when valid).** Conditioned on $z$, taking expectation over $\hat{x}_{1|t}^i \sim p_{1|t}^\theta(\cdot \mid z)$ yields

$$\mathbb{E}_{\hat{x}_{1|t}^i \mid z} \left[ q_t^{\mathrm{rvf}}(x^i \mid z, \hat{x}_{1|t}^i) \right] = q_t^{\mathrm{vf}}(x^i \mid z), \tag{20}$$

since $\mathbb{E}[\delta_{\hat{x}_{1|t}^i}(x^i) \mid z] = p_{1|t}^\theta(x^i \mid z)$. Therefore, in the regime where $1 - \lambda_t \geq 0$, rvf and vf induce the same Euler one-step transitional kernel.

**Clipping-and-renormalization safeguard.** When the Euler mixture produces negative entries (typically because $1 - \lambda_t < 0$), we apply the common safeguard *clip negatives and renormalize*:

$$\mathrm{CR}(q) := \frac{\max(0, q)}{\sum_{x^i} \max(0, q(x^i))}, \tag{21}$$

where $\max(0, \cdot)$ is elementwise. We now show that, for rvf, this safeguard exactly matches the ESP behavior in the endpoint regime.

**Lemma D.1** (rvf with clipping implements ESP when $1 - \lambda_t < 0$)**.** *Fix $z$ and a sampled candidate $\hat{x}^i_{1|t} \neq z^i$. Applying clipping-and-renormalization CR to the sparse rvf kernel in Eq.* (19) *gives*

$$\mathrm{CR}\Big(q^{\mathrm{rvf}}_t(\cdot \mid z, \hat{x}^i_{1|t})\Big) = \begin{cases} q^{\mathrm{rvf}}_t(\cdot \mid z, \hat{x}^i_{1|t}), & 1 - \lambda_t \geq 0, \\ \delta_{\hat{x}^i_{1|t}}(\cdot), & 1 - \lambda_t < 0. \end{cases} \tag{22}$$

*Proof.* If $1 - \lambda_t \geq 0$, both coefficients $(1 - \lambda_t)$ and $\lambda_t$ are nonnegative and sum to 1, so CR leaves the distribution unchanged. If $1 - \lambda_t < 0$, then the negative mass on $\delta_{z^i}$ is clipped to 0, leaving only the positive mass $\lambda_t$ on $\hat{x}^i_{1|t}$, which renormalizes to $\delta_{\hat{x}^i_{1|t}}$. $\qquad\square$

**Corollary D.2** (Endpoint behavior equals ESP)**.** *In the endpoint window where $1 - \lambda_t < 0$ may occur (cf. Theorem 3.1), rvf with clipping-and-renormalization reduces to the ESP endpoint rule: it directly samples from the model posterior:*

$$x^i_{t+\Delta t} \sim p^\theta_{1|t}(\cdot \mid z), \qquad \textit{equivalently, } x^i_{t+\Delta t} = \hat{x}^i_{1|t} \textit{ with } \hat{x}^i_{1|t} \sim p^\theta_{1|t}(\cdot \mid z). \tag{23}$$

*Thus, rvf can be viewed as implementing ESP implicitly in the endpoint regime while matching the vf Euler kernel in expectation when $1 - \lambda_t \geq 0$.*

## E. Experimental Details

### E.1. Computing Environment

Our implementation is based on PyG (Fey & Lenssen, 2019). The experiments are conducted on a single workstation with 8 A100 GPUs.

### E.2. Implementation Details and Reproducibility

We adopt the Graph Transformer backbone from DiGress (Vignac et al., 2022), with further experimental details available in our source code at https://github.com/LUOyk1999/SimGFM.

#### E.2.1. SPECIFICATION OF SOURCE DISTRIBUTION $p_0$

To ensure full reproducibility, we explicitly specify the source distribution $p_0$ used for initialization in each experiment. The choice of $p_0$ defines the prior noise distribution from which the backward generation process starts ($x_1 \sim p_0$).

*Table 4.* Source distribution ($p_0$) configurations for all datasets.

| Dataset | Node Distribution ($p_0^V$) | Edge Distribution ($p_0^E$) | Remarks |
|---|---|---|---|
| QM9 | Marginal | Marginal | — |
| QM9H | Marginal | Marginal | — |
| Planar | Marginal | Marginal | — |
| Tree | Marginal | Marginal | — |
| MOSES | Marginal | Marginal | — |
| Ego-Small | Marginal | Marginal | — |
| Community-Small | Marginal | Marginal | — |
| Grid | Marginal | Marginal | — |
| TLS | Marginal | Marginal | — |
| SBM | AbsorbFirst | AbsorbFirst | Initialized with absorbing state |

### E.2.2. SELECTED SCHEDULER ORDER $k$

For reproducibility, we list the default scheduler order $k$ used for each dataset in the main experiments, together with any step-specific exception noted in the remarks.

*Table 5.* Selected scheduler order $k$ for each dataset, with step-specific exceptions noted when applicable.

| Dataset | Selected $k$ | Remarks |
|---|---|---|
| QM9 | 20 | — |
| QM9H | 10 | — |
| Guacamol | 10 | — |
| MOSES | 10 | 500-step uses $k = 20$ |
| Tree | 10 | — |
| Planar | 10 | — |
| SBM | 3 | — |
| TLS | 10 | — |
| Ego-Small | 10 | — |
| Community-Small | 10 | — |
| Grid | 10 | — |

### E.2.3. SENSITIVITY TO SOURCE PRIORS

Our default initialization uses the Marginal prior on all datasets except SBM, where we use an absorbing-state prior. Figure 5 compares six source distributions on QM9 at 10 sampling steps: Uniform, Marginal, Node-Marginal, Edge-Marginal, Absorbing, and Masking. We observe that Marginal, Uniform, Node-Marginal, and Edge-Marginal all converge to similarly high validity, whereas Absorbing and especially Masking remain substantially worse throughout training. This supports our default use of the Marginal prior in the molecular setting.

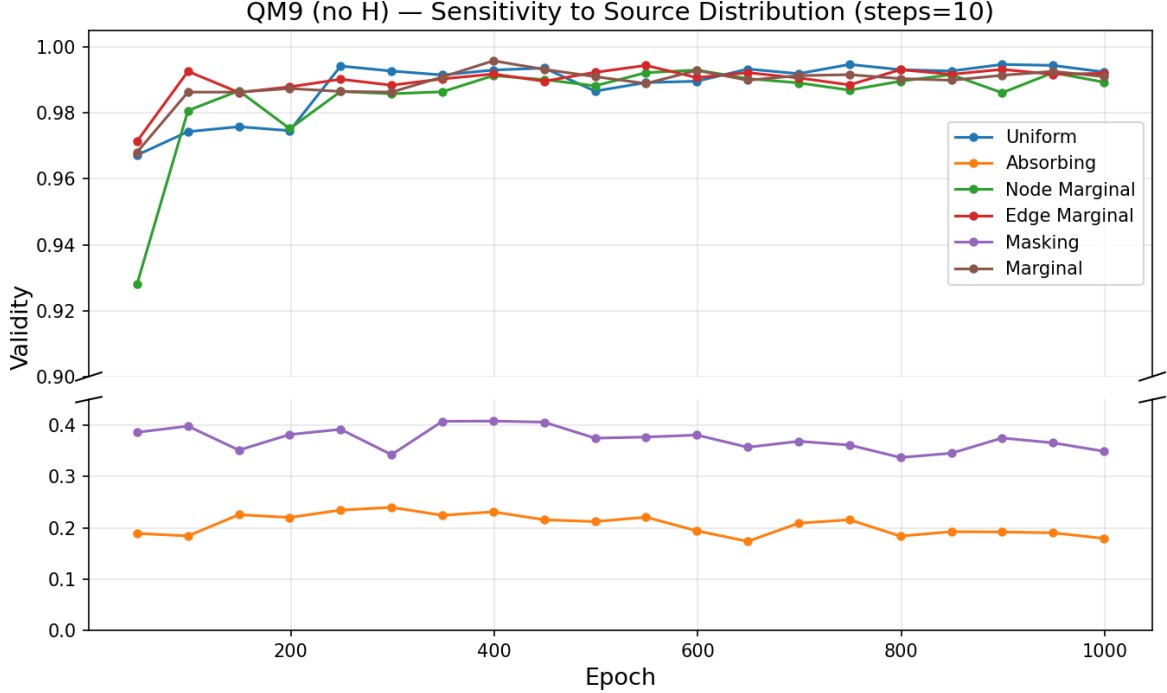

*Figure 5.* Sensitivity to source priors on QM9 at 10 sampling steps.

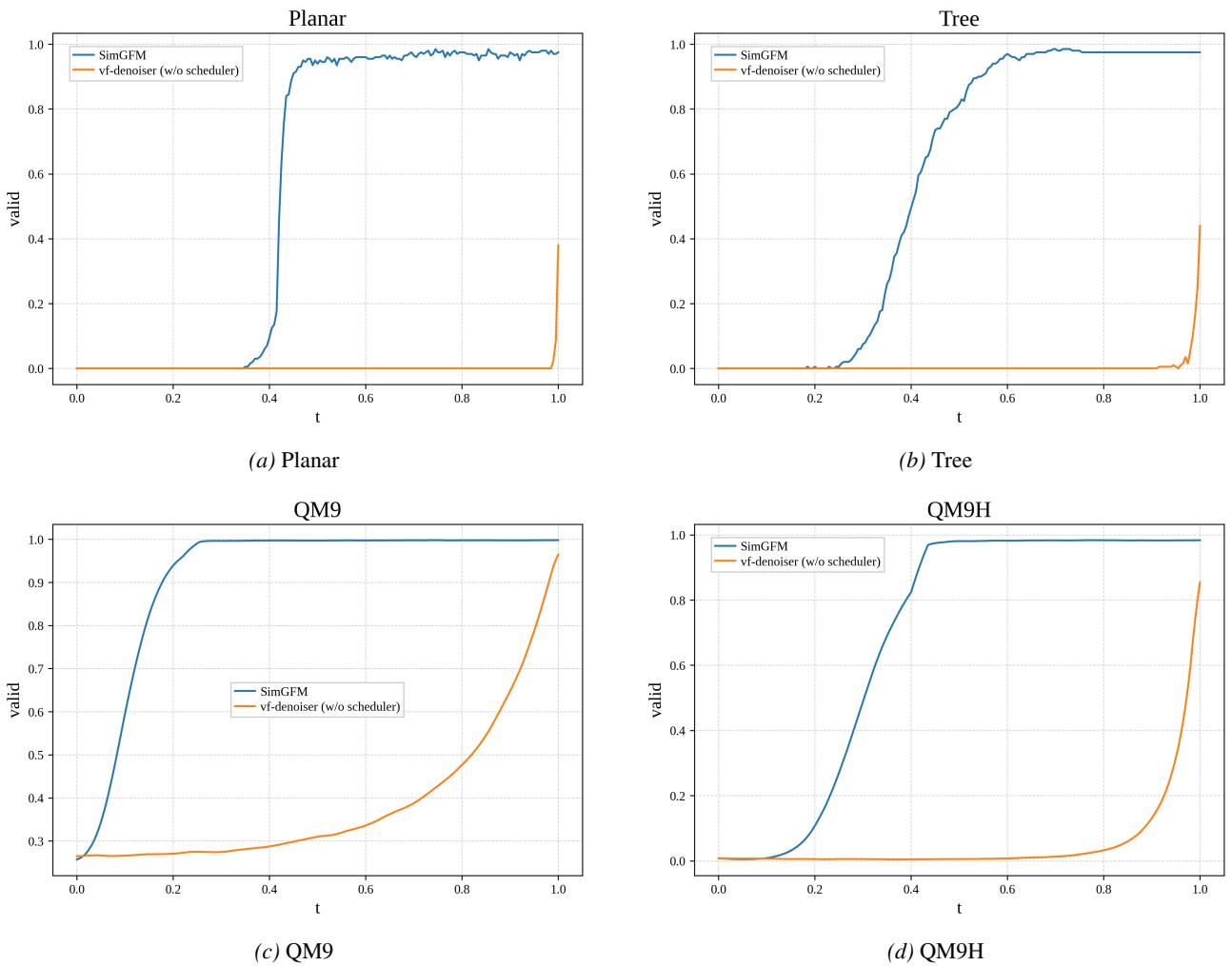

*(a)* Planar

*(b)* Tree

*(c)* QM9

*(d)* QM9H

*Figure 6.* Validity vs. time $t$ on four datasets.

## F. Further Results

### F.1. Validity Emergence along the Denoising Trajectory

**Discussion.** Figure 6 visualizes how validity evolves over continuous time $t$ during sampling. Across datasets, uniform schedules tend to postpone most validity gains to the endpoint regime ($t \to 1$), producing a sharp late-stage rise. In contrast, the polynomial scheduler redistributes computation to the late refinement region and leads to an earlier and more stable increase in validity, consistent with our motivation in Sec. 3.2.

### F.2. Supplementary Analysis: Heuristic-Enriched vs. Pure DFM

As shown in Figure 7, under a sufficiently large sampling budget (500-1000 steps), heuristic-enriched SOTA variants (QIN et al., 2025; Hou et al., 2025) do not consistently outperform pure DFM on some benchmarks.

### F.3. Comparison on Ego-small, Community-small, and Grid Datasets

Following the GGFlow (Hou et al., 2025) setting, we use a 4:1 train/test split and report the mean over three runs. Table 6 shows that SimGFM with **200** steps achieves consistently small deviations across degree, clustering, and orbit statistics, reaching or approaching the best overall scores among all compared methods.

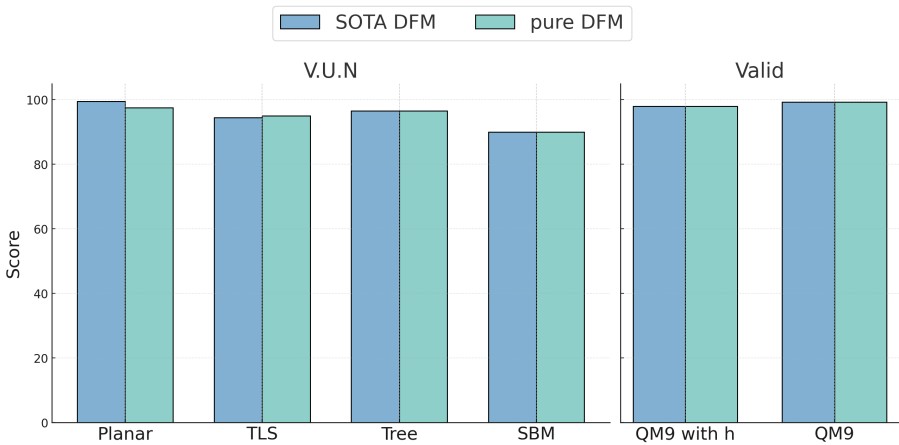

*Figure 7.* Comparison of ablation study on the QM9 dataset.

*Table 6.* Generation results on the generic graph datasets. Results are the means of 3 different runs. The best results and the second-best results are marked **bold** and underline.

| Model | # Steps↓ | Ego-small | | | | Community-small | | | | Grid | | | |
|---|---|---|---|---|---|---|---|---|---|---|---|---|---|
| | | Deg.↓ | Clus.↓ | Orbit↓ | Avg.↓ | Deg.↓ | Clus.↓ | Orbit↓ | Avg.↓ | Deg.↓ | Clus.↓ | Orbit↓ | Avg.↓ |
| Training Set | - | 0.014 | 0.022 | 0.004 | 0.013 | 0.003 | 0.009 | 0.001 | 0.005 | 0.000 | 0.000 | 0.000 | 0.000 |
| GraphRNN | - | 0.090 | 0.220 | 0.003 | 0.104 | 0.080 | 0.120 | 0.040 | 0.080 | 0.064 | 0.043 | 0.021 | 0.043 |
| EDP-GNN | 1000 | 0.054 | 0.092 | 0.007 | 0.051 | 0.050 | 0.159 | 0.027 | 0.079 | 0.460 | 0.243 | 0.316 | 0.340 |
| GDSS | 1000 | 0.027 | 0.033 | 0.008 | 0.022 | 0.044 | 0.098 | 0.009 | 0.058 | 0.133 | 0.009 | 0.123 | 0.088 |
| DiGress | 500 | 0.028 | 0.046 | 0.008 | 0.027 | 0.032 | 0.047 | 0.009 | 0.025 | 0.037 | 0.046 | 0.069 | 0.051 |
| GGFlow | 500 | 0.005 | 0.033 | 0.004 | 0.014 | **0.011** | 0.030 | **0.002** | **0.014** | 0.030 | **0.000** | 0.016 | 0.015 |
| CatFlow | - | 0.013 | 0.024 | 0.008 | 0.015 | 0.018 | 0.086 | 0.007 | 0.037 | 0.115 | 0.004 | 0.075 | 0.065 |
| DeFoG (50 steps) | **50** | 0.034 | 0.012 | 0.067 | 0.039 | 0.029 | 0.157 | 0.052 | 0.079 | 0.004 | **0.000** | **0.000** | 0.001 |
| DeFoG (200 steps) | 200 | 0.056 | 0.149 | 0.068 | 0.091 | 0.022 | 0.040 | **0.002** | 0.022 | 0.001 | **0.000** | **0.000** | **0.000** |
| SimGFM (50 steps) | **50** | **0.004** | 0.024 | 0.006 | 0.011 | 0.038 | 0.081 | 0.008 | 0.043 | **0.000** | **0.000** | **0.000** | **0.000** |
| SimGFM (200 steps) | 200 | 0.006 | **0.009** | **0.001** | **0.005** | 0.031 | **0.027** | **0.002** | 0.020 | **0.000** | **0.000** | **0.000** | **0.000** |

## F.4. Comparison with SMILES-Based and Fragment-Based Molecular Generators

To position SimGFM in a broader molecular-generation context, we additionally compare against representative SMILES-based and fragment-based baselines on GuacaMol and MOSES; the non-graph baseline numbers are taken from the comparison table reported by DeFoG (QIN et al., 2025).

*Table 7.* Comparison with representative SMILES-based and graph-based molecular generators on GuacaMol.

| Model | Class | Val. ↑ | V.U.N. ↑ | KL div ↑ | FCD ↑ |
|---|---|---|---|---|---|
| LSTM | SMILES | 95.9 | 87.4 | 99.1 | 91.3 |
| DiGress | Graph | 85.2 | 85.1 | 92.9 | 68.0 |
| DisCo | Graph | 86.6 | 86.5 | 92.6 | 59.7 |
| Cometh | Graph | 98.9 | 97.6 | 96.7 | 72.7 |
| DeFoG (500) | Graph | 99.0 | 97.9 | 97.7 | 73.8 |
| SimGFM (50) | Graph | 98.9 | 98.3 | 93.6 | 79.6 |
| SimGFM (200) | Graph | 99.2 | 98.8 | 94.0 | 80.8 |

*Table 8.* Comparison with representative SMILES-based, fragment-based, and graph-based molecular generators on MOSES.

| Model | Class | Val. ↑ | Filters ↑ | FCD ↓ | SNN ↑ | Scaf. ↑ |
|---|---|---|---|---|---|---|
| VAE | SMILES | 97.7 | 99.7 | 0.57 | 0.58 | 5.9 |
| JT-VAE | Fragment | 100.0 | 97.8 | 1.00 | 0.53 | 10.0 |
| DiGress | Graph | 85.7 | 97.1 | 1.19 | 0.52 | 14.8 |
| DisCo | Graph | 88.3 | 95.6 | 1.44 | 0.50 | 15.1 |
| Cometh | Graph | 90.5 | 99.1 | 1.27 | 0.54 | 16.0 |
| DeFoG (500) | Graph | 92.8 | 98.9 | 1.95 | 0.55 | 14.4 |
| SimGFM (50) | Graph | 86.1 | 95.3 | 1.44 | 0.51 | 16.9 |
| SimGFM (200) | Graph | 89.4 | 98.9 | 1.08 | 0.55 | 17.3 |

These comparisons show that SimGFM is highly competitive within graph-based molecular generation: on GuacaMol, SimGFM (50) already improves over DeFoG (500) in FCD, and on MOSES, SimGFM (200) attains the best FCD among the compared graph-based baselines (QIN et al., 2025). At the same time, strong SMILES-based methods such as LSTM and VAE still achieve stronger distribution-matching scores on some metrics, which we regard as a broader limitation of graph-native generation rather than a limitation unique to SimGFM.

### F.5. Conditional Generation

We evaluate conditional generation on TLS dataset. Performance is assessed by (i) TLS Valid, measuring consistency between generated graphs and provided labels, and (ii) V.U.N. (validity, uniqueness, and novelty), where a graph is considered valid if it is both planar and connected. For fairness, We follow the same evaluation protocol as DeFoG and use their released evaluation code to compute both TLS Valid and V.U.N. we report the mean performance of existing methods on two subsets, as summarized in Table 9. SimGFM achieves **96.3**% TLS Valid and **96.3**% V.U.N. with only **200** steps, matching or surpassing DeFoG while requiring far fewer inference steps.

*Table 9.* TLS conditional generation results.

| Model | TLS Dataset | |
|---|---|---|
| | V.U.N. ↑ | TLS Val. ↑ |
| Train set | 0.0 | 100 |
| GraphGen | 40.2 | 25.1 |
| BiGG | 0.6 | 16.7 |
| SPECTRE | 7.9 | 25.3 |
| DiGress | 13.2 | 12.6 |
| ConStruct | **99.1** | 92.1 |
| DeFoG (50 steps) | 44.5 | 93.0 |
| DeFoG (1000 steps) | 94.5 | 95.8 |
| SimGFM (50 steps) | 81.3 | 91.3 |
| SimGFM (200 steps) | 96.3 | **96.3** |

### F.6. Ablation on ESP (Sec. 3.3)

*Table 10.* ESP ablation.

| Dataset | vf-denoiser | vf-denoiser+ESP | Gain |
|---|---|---|---|
| TLS | 2.50 | 96.25 | +93.75 |
| QM9-with-H | 97.25 | 98.40 | +1.15 |
| MOSES | 85.78 | 89.39 | +3.61 |

*Table 11.* Ablation on ESP and scheduler choice at 200 sampling steps. We compare the identity scheduler and our polynomial scheduler, with and without ESP. For QM9 and QM9H, we report both validity and FCD; for Tree and Planar, we report validity.

| Method | QM9 | | QM9H | | Tree | Planar |
|---|---|---|---|---|---|---|
| | **Valid** | **FCD** | **Valid** | **FCD** | **Valid** | **Valid** |
| vf-denoiser (w/ identity scheduler) | $98.3 \pm 0.2$ | $0.10 \pm 0.00$ | $97.7 \pm 0.1$ | $0.05 \pm 0.00$ | $49.5 \pm 4.0$ | $38.0 \pm 5.0$ |
| vf-denoiser+ESP (w/ identity scheduler) | $99.3 \pm 0.1$ | $0.10 \pm 0.00$ | $95.6 \pm 0.1$ | $0.05 \pm 0.00$ | $57.0 \pm 4.6$ | $57.5 \pm 6.5$ |
| vf-denoiser (w/ our scheduler) | $99.6 \pm 0.0$ | $0.11 \pm 0.00$ | $97.7 \pm 0.1$ | $0.04 \pm 0.00$ | $95.5 \pm 1.0$ | $96.0 \pm 3.0$ |
| vf-denoiser+ESP (w/ our scheduler) | $99.8 \pm 0.0$ | $0.15 \pm 0.00$ | $98.4 \pm 0.1$ | $0.04 \pm 0.00$ | $99.5 \pm 1.0$ | $100.0 \pm 0.0$ |

These results show that ESP has near-zero effect on FCD relative to the much larger changes induced by scheduler order, while substantially improving validity whenever the endpoint Euler step becomes unstable. In particular, on QM9 the FCD change introduced by ESP remains small compared with the increase caused by over-concentrating the scheduler.

**Implementation note on ESP.**   When ESP is disabled, and the identity scheduler is used, which corresponds to the linear case $f_k(t)$ with $k = 1$, we can observe that although this scheduler is substantially milder than high-$k$ polynomial schedulers, the Euler one-step mixture can still be numerically fragile in the endpoint region, since the last update may approach the boundary regime where the "stay" coefficient becomes small and the discretized kernel may require projection to preserve nonnegativity.

In some ablations, ESP is implemented following Sec. D.

## F.7. Sensitivity Analysis over Scheduler Order $k$

*Table 12.* Sensitivity to scheduler order $k$ on QM9 (10 steps). Results are mean $\pm$ std over 3 sampling runs of 10,000 generated graphs each.

| $k$ | Valid $\uparrow$ | Relaxed Valid $\uparrow$ | Unique $\uparrow$ | FCD $\downarrow$ |
|---|---|---|---|---|
| 1 | 75.8$\pm$0.6 | 86.3$\pm$0.2 | 96.8$\pm$0.1 | 0.68$\pm$0.01 |
| 2 | 95.8$\pm$0.2 | 97.8$\pm$0.2 | 96.8$\pm$0.1 | 0.22$\pm$0.01 |
| 5 | 99.3$\pm$0.1 | 99.6$\pm$0.1 | 96.4$\pm$0.1 | 0.26$\pm$0.00 |
| 10 | 99.4$\pm$0.1 | 99.6$\pm$0.1 | 95.5$\pm$0.1 | 0.55$\pm$0.01 |
| 20 | 99.5$\pm$0.0 | 99.7$\pm$0.0 | 95.0$\pm$0.2 | 0.92$\pm$0.00 |
| 30 | 99.5$\pm$0.0 | 99.7$\pm$0.0 | 94.6$\pm$0.2 | 1.17$\pm$0.03 |

*Table 13.* Sensitivity to scheduler order $k$ on QM9 (200 steps). Results are mean $\pm$ std over 3 sampling runs of 10,000 generated graphs each.

| $k$ | Valid $\uparrow$ | Relaxed Valid $\uparrow$ | Unique $\uparrow$ | FCD $\downarrow$ |
|---|---|---|---|---|
| 1 | 99.3$\pm$0.1 | 99.6$\pm$0.1 | 96.7$\pm$0.0 | 0.10$\pm$0.00 |
| 2 | 99.7$\pm$0.0 | 99.8$\pm$0.1 | 96.5$\pm$0.2 | 0.10$\pm$0.01 |
| 5 | 99.8$\pm$0.1 | 99.8$\pm$0.1 | 96.2$\pm$0.1 | 0.11$\pm$0.00 |
| 10 | 99.8$\pm$0.0 | 99.9$\pm$0.0 | 96.2$\pm$0.1 | 0.13$\pm$0.01 |
| 20 | 99.8$\pm$0.0 | 99.8$\pm$0.0 | 95.9$\pm$0.0 | 0.15$\pm$0.00 |
| 30 | 99.8$\pm$0.0 | 99.9$\pm$0.0 | 95.7$\pm$0.2 | 0.16$\pm$0.00 |

*Table 14.* Sensitivity to scheduler order $k$ on Planar and Tree (200 steps). V.U.N. denotes Valid, Unique, and Novel. Values are mean $\pm$ std from five runs of 40 graphs each.

| $k$ | Planar | | Tree | |
|---|---|---|---|---|
| | V.U.N. $\uparrow$ | Ratio $\downarrow$ | V.U.N. $\uparrow$ | Ratio $\downarrow$ |
| 1 | 57.5$\pm$6.5 | 2.3$\pm$1.2 | 57.0$\pm$4.6 | 1.8$\pm$0.6 |
| 2 | 97.0$\pm$3.7 | 1.6$\pm$0.9 | 95.0$\pm$3.2 | 2.0$\pm$0.7 |
| 5 | 99.5$\pm$1.0 | 1.6$\pm$0.5 | 98.0$\pm$1.9 | 2.1$\pm$0.8 |
| 10 | 100.0$\pm$0.0 | 9.3$\pm$2.6 | 99.5$\pm$1.0 | 1.5$\pm$0.2 |
| 20 | 100.0$\pm$0.0 | 8.2$\pm$1.3 | 97.5$\pm$1.6 | 2.8$\pm$1.4 |
| 30 | 99.5$\pm$1.0 | 15.4$\pm$2.7 | 97.5$\pm$2.2 | 3.8$\pm$1.0 |

*Table 15.* Sensitivity to scheduler order $k$ on Planar and Tree (1000 steps). V.U.N. denotes Valid, Unique, and Novel. Values are mean $\pm$ std from five runs of 40 graphs each.

| $k$ | Planar | | Tree | |
|---|---|---|---|---|
| | V.U.N. $\uparrow$ | Ratio $\downarrow$ | V.U.N. $\uparrow$ | Ratio $\downarrow$ |
| 1 | 93.5$\pm$4.9 | 1.4$\pm$0.3 | 87.0$\pm$1.0 | 2.1$\pm$1.0 |
| 2 | 100.0$\pm$0.0 | 2.2$\pm$0.4 | 97.5$\pm$2.7 | 1.8$\pm$0.6 |
| 3 | 100.0$\pm$0.0 | 7.9$\pm$0.9 | 96.5$\pm$1.2 | 1.9$\pm$0.9 |
| 5 | 100.0$\pm$0.0 | 19.9$\pm$2.6 | 99.0$\pm$1.2 | 2.7$\pm$0.9 |
| 10 | 100.0$\pm$0.0 | 21.0$\pm$1.5 | 100.0$\pm$0.0 | 2.1$\pm$0.8 |

## F.8. Discussion of the Planar Validity–Ratio Trade-off

As analyzed in Sec. 3.2, sufficiently many iterations near the endpoint ($t \rightarrow 1$) help refine invalid structures into valid ones, thereby improving validity. However, once validity is already near saturation, continuing dense endpoint updates can hurt diversity: the model prediction $p_{1|t}^{\theta}$ receives dominant weight in the update and tends to concentrate probability mass on frequent structures, gradually overwriting low-frequency yet still valid ones. Since Ratio measures how well the generated structural distribution matches the training distribution, losing these valid low-frequency modes naturally increases Ratio.

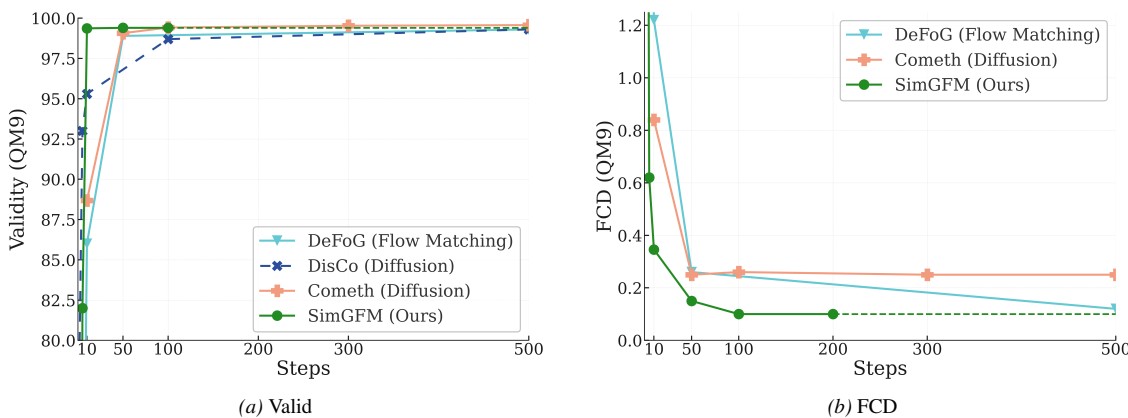

*Figure 8.* Sampling Efficiency on QM9

We conducted a dedicated $k$-sensitivity analysis at 200 and 1000 steps; see Tables 14 and 15. This sensitivity analysis supports the above interpretation. At 200 steps, $k = 5$ significantly improves Ratio over $k = 10$ with almost no loss in validity. At 1000 steps, $k = 2$ likewise performs better than larger $k$ values. QM9 at 10 steps shows the same pattern: as $k$ increases, the gain in validity saturates while FCD degrades rapidly.

We also note that the MMD-based Ratio metric can fluctuate under small-sample evaluation (O'Bray et al., 2021; Krimmel et al., 2025).

## G. Further Discussion

### G.1. Current Evaluation Limitations

Our experiments follow standard evaluation protocols from prior graph-generation work, but several limitations should be noted. First, for synthetic graph benchmarks, we report MMD-style statistics and the derived Ratio score for fair comparison with prior work. These metrics are widely used, but they are not fully robust absolute measures and can be sensitive to kernel choice, sample size, and estimator variance (O'Bray et al., 2021).

Second, classifier-based alternatives such as PolyGraph Discrepancy (PGD) may provide a stronger complementary view of graph generation quality than kernel-based distances alone (Krimmel et al., 2025).

Third, some standard synthetic benchmarks such as Planar, SBM, and Tree are relatively small, so metric estimates may still carry non-negligible variance. Larger benchmark variants may therefore provide a cleaner setting for future evaluation (Krimmel et al., 2025).

### G.2. Graph Generation in Hyperbolic Space

Many real-world graphs exhibit hierarchical, tree-like, or non-Euclidean structural patterns, suggesting that the geometry of the generative space may affect the model's ability to capture structural relations. Existing hyperbolic or Riemannian graph-generation methods, such as HGDM, HypDiff, and GeoMancer, exploit this observation by introducing continuous manifold geometry into graph generation, typically through latent diffusion in hyperbolic or Riemannian spaces (Wen et al., 2024; Fu et al., 2024; Gao et al., 2025).

For SimGFM, we see two possible directions for incorporating such geometric inductive biases. The first is to improve the neural parameterization of the discrete transition model with geometry-aware components. For example, hyperbolic graph encoders, manifold-aware attention layers, geometry-aware category embeddings, or structure-conditioned rate parameterizations could be used to better capture hierarchical graph structure. This direction is naturally compatible with SimGFM, since the sampler can still evolve categorical node and edge distributions on the probability simplex, while the hyperbolic geometry enters the model through the denoiser or transition parameterization.

A more fundamental direction is to construct the generative velocity field itself in hyperbolic or Riemannian space. Developing such a framework would require additional mathematical theory, and we view it as a potential direction for

future work.

## G.3. LLM Usage

We used large language models (LLMs) for language editing and polishing only.

## G.4. Ethics Statement

Our study does not involve human subjects, sensitive personal data, or applications with foreseeable harmful impact. All datasets used are publicly available, and we follow community standards regarding data usage, fairness, and privacy.

