# OpenReview forum: "SimGFM: Simplifying Discrete Flow Matching for Graph Generation"
_ICML.cc/2026/Conference — ICML 2026 regular_

### Official Review · Reviewer_1abu · 2026-03-08

**Soundness:** 3
**Presentation:** 3
**Significance:** 3
**Originality:** 3
**Overall Recommendation:** 4
**Confidence:** 4

**Summary:**

The paper proposes SimGFM, a simplified discrete flow matching framework for graph generation. SimGFM shows improved performance with fewer steps on QM9 and good performance on several commonly used benchmarks.

**Compliance With Llm Reviewing Policy:**

Affirmed.

**Final Justification:**

The rebuttal has addressed all my concerns, and I will keep my positive score.

**Key Questions For Authors:**

- Is the proposed framework applicable to other discrete data beyond graphs, such as biological sequences or texts?

**Limitations:**

The limitations of this work are not systematically discussed. I think the authors should at least discuss the limitations of the synthetic datasets and evaluation metrics, which have commonly known issues [1,2].

[1]: O'Bray, Leslie, et al. Evaluation Metrics for Graph Generative Models: Problems, Pitfalls, and Practical Solutions. ICLR 2022.

[2]: Krimmel, Markus, et al. PolyGraph Discrepancy: a classifier-based metric for graph generation. ICLR 2026.

**Strengths And Weaknesses:**

### Strengths
- The paper is well written and easy to follow.
- The motivation is clear, the approach is technically sound, and the experiments are thorough.

### Weaknesses
Probably not a weakness specific to SimGFM, but in general, graph flow matching models do not perform as well as SMILES-based models. For example, LSTM achieves 91.3 in terms FCD on GuacaMol from the original GuacaMol paper. I recommend the authors explicitly add the state-of-the-art models on both GuacaMol and MOSES benchmarks and properly discuss the advantages and limitations of SimGFM within this broader context.

---

> ### Author Rebuttal · Authors · 2026-03-31
>
> Thank you for the positive evaluation. We address your suggestion on SMILES-based comparisons, broader applicability, and limitations.
>
> **(W1-1) Comparison with SMILES-based models.** We added representative comparisons on both GuacaMol and MOSES (baseline results cited from DeFoG\[3\]).
>
> **GuacaMol**
>
> |Model|Class|Val.|V.U.N.|KL div|FCD|
> |-|-|-:|-:|-:|-:|
> |LSTM|SMILES|95.9|87.4|**99.1**|**91.3**|
> |DiGress|Graph|85.2|85.1|92.9|68.0|
> |DisCo|Graph|86.6|86.5|92.6|59.7|
> |Cometh|Graph|98.9|97.6|96.7|72.7|
> |DeFoG (500)|Graph|99.0|97.9|97.7|73.8|
> |**SimGFM (50)**|**Graph**|98.9|98.3|93.6|79.6|
> |**SimGFM (200)**|**Graph**|**99.2**|**98.8**|94.0|**80.8**|
>
> **MOSES**
>
> |Model|Class|Val.|Filters|FCD|SNN|Scaf.|
> |-|-|-:|-:|-:|-:|-:|
> |VAE|SMILES|97.7|**99.7**|**0.57**|**0.58**|5.9|
> |JT-VAE|Fragment|**100.0**|97.8|1.00|0.53|10.0|
> |DiGress|Graph|85.7|97.1|1.19|0.52|14.8|
> |DisCo|Graph|88.3|95.6|1.44|0.50|15.1|
> |Cometh|Graph|90.5|99.1|1.27|0.54|16.0|
> |DeFoG (500)|Graph|92.8|98.9|1.95|0.55|14.4|
> |**SimGFM (50)**|**Graph**|86.1|95.3|1.44|0.51|16.9|
> |**SimGFM (200)**|**Graph**|89.4|98.9|**1.08**|0.55|17.3|
>
> These results show that **within graph-based generation, SimGFM is highly competitive**: `SimGFM (50)` already surpasses `DeFoG (500)` on GuacaMol FCD (`79.6` vs. `73.8`), and `SimGFM (200)` reaches the best MOSES FCD (`1.08`) among graph-based baselines. At the same time, strong **SMILES-based** methods such as LSTM/VAE still perform better on distribution-matching metrics, which we view as a broader limitation of graph generation.
>
> **(W1-2) Advantages and limitations of SimGFM in this broader context.** SimGFM preserves the flexibility of graph-native generation while greatly improving sampling efficiency: on many datasets, `10-50` steps are enough to match or exceed methods that previously needed `500-1000` steps. Its main limitation is that graph-based methods still trail the best SMILES-based models on distribution-matching metrics, likely because SMILES better encodes valence rules and aligns with the ChemNet encoder used by FCD.
>
> **(Q1) Applicability beyond graphs.** We believe the core design principle of SimGFM is potentially transferable beyond graphs. The essential idea is **endpoint-focused allocation of sampling budget**: if a discrete generation task exhibits the same phenomenon that final decisions near $t \to 1$ dominate sample quality, then the same scheduler and ESP logic should be beneficial. This could plausibly apply to biological sequences or text, but we do not want to overclaim: whether the same endpoint-sensitive pattern holds must be verified empirically for each domain. So our stronger claim is that SimGFM provides a **general methodology** for analyzing discrete generation trajectories and designing more efficient samplers around their critical phases.
>
> **(L1) Limitations.** We fully agree that the limitations should be discussed more systematically. We will add the following points to the revision. First, our evaluation on synthetic graph benchmarks follows standard MMD-style metrics and their derived Ratio scores for fair comparison with prior work, but these metrics have known issues: they can be sensitive to kernel choice and may not provide a fully robust absolute ranking [1]. Second, recently proposed alternatives such as PolyGraph Discrepancy (PGD) [2] may offer a stronger classifier-based evaluation of graph generation quality. Third, some synthetic datasets such as Planar/SBM/Tree are relatively small, so metric estimates may have non-negligible variance; Krimmel et al. [2] accordingly propose larger-scale benchmarks (Planar-L, SBM-L, Lobster-L) to reduce estimation bias. We agree that stronger future evaluations should combine these larger benchmarks with more robust metrics such as PGD.
>
> We appreciate these suggestions and will incorporate them in the revision.
>
> ---
>
> [1] Evaluation Metrics for Graph Generative Models: Problems, Pitfalls, and Practical Solutions. ICLR, 2022.
>
> [2] PolyGraph Discrepancy: A Classifier-Based Metric for Graph Generation. ICLR, 2026.
>
> [3] DeFoG: Discrete Flow Matching for Graph Generation. ICML, 2025.

---

> > ### Author Rebuttal · Reviewer_1abu · 2026-04-02
> >
> > I thank the authors for their detailed rebuttal, which has addressed all my concerns. I will keep my positive score.

---

> > > ### Author Response · Authors · 2026-04-02
> > >
> > > Dear Reviewer 1abu,
> > >
> > > Thank you very much for your time and thoughtful follow-up. We sincerely appreciate your recognition that our rebuttal has addressed your concerns, as well as your positive evaluation of our work.
> > >
> > > Best regards,
> > >
> > > The Authors

---

### Official Review · Reviewer_TfNh · 2026-03-08

**Soundness:** 3
**Presentation:** 3
**Significance:** 2
**Originality:** 2
**Overall Recommendation:** 4
**Confidence:** 3

**Summary:**

The authors proposed a Discrete Flow Matching (DFM) framework for graph generation designed to overcome the high complexity and slow sampling speeds of existing models. The proposed method leverages the DFM theory and drastically reduces the required steps, which includes the use of an rate matrix based on the vf-denoiser, a polynomial scheduler that allocates more computational budget and denser updates near the endpoint  formulation, and the use of ESP to act as a safeguard by replacing the standard update.

**Compliance With Llm Reviewing Policy:**

Affirmed.

**Key Questions For Authors:**

n/a

**Limitations:**

yes

**Strengths And Weaknesses:**

Strength:
- The authors introduces Discrete Flow Matching (DFM) based approach for graph generation, which some insightful observations. The result significantly increases the efficiency of graph generation.

- The overall technical proposal is basically sound and clear. For example, the authors also provide a solid mathematical basis (Theorem 3.1) identifying exactly when Euler discretization causes negative probability masses, justifying the need for a safeguard. The paper effectively communicates its core motivation through clear, empirical trajectory-level observations.

- In terms of significance, this is a great improvement on the efficiency by leveraging the merits of flow matching, and it can achieve significant improvement over existing methods while maintains reasonable accuracy. This is also justifies the novelty of the work from the interdisciplinary area of flow matching and graph generalization.

Weakness:
- There is some technical vagueness, despite the overall presentation is clear and easy to follow. For example, there is some discussion on the Endpoint Safety Projection (ESP, but its theoretical grounding is slightly unclear. The authors argue that replacing the standard DFM update with direct posterior sampling is a "reasonable endpoint approximation”, but not very convincing. A bit more details or insights may help.

- Diffusion model or Flow matching are well know for generation, although the graph generation is a new topic. The framework is essentially a synthesis of existing techniques rather than a fundamentally new generative paradigm. The core velocity-field formulation is directly adopted from the previously established vf-denoiser. The authors should clarify the true novelty contributions.

- In Section 4.4 and Appendix F.5 (Table 8) the authors said that even with a mild identity scheduler, the update enters a boundary regime where the stay coefficient becomes small and the discretized kernel may require projection to preserve nonnegativity. This is a bit vague and triggers concern on the stability. The author should provide a bit more details.

- The authors did not discuss the impact of the space where the velocity field is built. For example, it is known for many graph based work, the hyperbolic space helps represent structural relationships and some hierarchy relationship over a graph. The authors should look into that and propose some discussion.

- In Tables 2 and 3, several experimental results achieved a large range, such as the Unique, varying from 2.0 to 100. The authors should present some insights on that, if it is related to how strong the baseline is or any parameter tuning.

---

> ### Author Rebuttal · Authors · 2026-03-31
>
> Thank you for the positive review. We address each concern below.
>
> **(W1) Theoretical grounding of ESP.** We agree this point deserves a clearer explanation. Under vf-denoiser, the Euler update can be written as
>
> $
> p_{t+h}(x) = \lambda_t  p_{1|t}(x) + (1-\lambda_t) \delta_t(x), \quad \text{where} \quad \lambda_t = \frac{h \, \dot{\kappa}_t}{1-\kappa_t}.
> $
>
> This is a convex combination only when $\lambda_t \in [0,1]$. Once $\lambda_t > 1$, the “stay” coefficient $1-\lambda_t$ becomes negative, which creates the invalid probability mass identified in Theorem 3.1. ESP can therefore be interpreted as replacing $\lambda_t$ by ${\lambda_t}^{\prime} = \text{min}(\lambda_t, 1)$. Concretely, at time points where $\lambda_t > 1$, we retain $\kappa_t^{\prime} = \kappa_t$ but replace the derivative by
>
> $\dot{{\kappa_t}^{\prime}} = \frac{1-{\kappa_t}^{\prime}}{h},$
>
> which yields a corrected scheduler ${\kappa_t}^{\prime}$ on the discrete time grid $t_n = nh$. This ${\kappa_t}^{\prime}$ can always be constructed to satisfy all requirements of [5]: $\kappa_0=0$, $\kappa_1=1$, monotonic, and continuously differentiable. Moreover, the monotonic growth of $\lambda_t$ embodies a natural inductive bias: as $t$ increases, $p_{1|t}$ should contribute more to the update; clamping it to 1 preserves this trend while avoiding negative probabilities. Thus ESP is not a heuristic endpoint approximation but a theoretically consistent scheduler correction within the DFM framework.
>
> **(W2) What is truly novel in SimGFM?** We agree that vf-denoiser itself is not our claimed contribution. Our novelty lies in what we build on top of it:
>
> 1. **A new empirical observation:** validity changes most sharply near the endpoint in graph generation trajectories, a pattern not previously systematically analyzed or exploited.
> 2. **A theoretical result:** Theorem 3.1 quantifies exactly when Euler discretization under vf-denoiser produces negative probabilities, linking scheduler order, step size, and instability.
> 3. **A principled remedy:** ESP is shown to be equivalent to a scheduler correction rather than an external patch.
> 4. **A practical outcome:** the resulting minimal design reaches or exceeds prior `500-1000`-step baselines on 10 datasets using only `10-50` steps, while shrinking the search space from DeFoG's `omega x eta` heuristics to a single `k`.
>
> So the contribution is not a new paradigm, but an analysis-driven design principle for efficient graph DFM.
>
>
> **(W3) Stability in the boundary regime.** For the identity scheduler $\kappa_t=t$, the stay coefficient becomes $1 - \frac{h}{1-t}$. On the final uniform-grid step, $1-t = h$, so this coefficient is mathematically zero. In practice, finite-precision arithmetic can make it slightly negative, so even a mild scheduler may need projection at the boundary. In this sense, “projection” at `k=1` is already a common engineering special case: one simply outputs the one-shot prediction $p_{1|t}(x)$ at the final step. This is a special case of ESP that clips $\lambda_t$ only at the last step. SimGFM's ESP generalizes this to high-order schedulers (`k>1`), where $\lambda_t > 1$ is not a rare floating-point accident but **mathematically inevitable** in the tail window, requiring correction across multiple final steps.
>
> **(W4) Hyperbolic space.** We appreciate this suggestion. Existing hyperbolic graph generation methods (e.g., HGDM [2], HypDiff [3], GeoMancer [4]) follow a latent diffusion route, encoding graphs into a continuous hyperbolic/Riemannian latent space. SimGFM, by contrast, is a **discrete flow-matching** method defined directly on categorical node/edge distributions over the probability simplex, so “which geometry the velocity field lives in” has a different meaning. That said, hyperbolic geometry could still benefit DFM, e.g., through hyperbolic graph attention layers or category embeddings for structure-aware transition kernels. We will add this discussion.
>
> **(W5) Large numeric ranges in Tables 2 and 3.** We suspect the comment refers to **FCD** rather than Unique. The apparent range comes from benchmark protocol differences: QM9/QM9H use the `fcd` library to compute raw FCD distances (lower is better), MOSES uses the `moses` library similarly (lower is better), whereas GuacaMol uses the official `guacamol` library, which normalizes FCD into a `[0,100]` score (higher is better [1]). The discrepancy reflects evaluation conventions and dataset complexity, not method instability. If we misunderstood the comment, we welcome clarification.
>
> We will revise the paper to sharpen both theory and presentation.
>
> ---
>
> [1] GuacaMol: Benchmarking Models for de Novo Molecular Design. JCIM, 2019.
>
> [2] Hyperbolic Graph Diffusion Model. AAAI, 2024.
>
> [3] Hyperbolic Geometric Latent Diffusion Model for Graph Generation. ICML, 2024.
>
> [4] Toward a Unified Geometry Understanding: Riemannian Diffusion Framework for Graph Generation and Prediction. NeurIPS, 2025.
>
> [5] Discrete Flow Matching. NeurIPS, 2024.

---

> > ### Author Rebuttal · Reviewer_TfNh · 2026-04-06
> >
> > We thank the authors for their clear and thorough rebuttal. Therefore, we are upholding our positive recommendation.

---

> > > ### Author Response · Authors · 2026-04-06
> > >
> > > Dear Reviewer TfNh,
> > >
> > > Thank you for taking the time to carefully review our rebuttal and for confirming that your concerns have been adequately addressed. We will incorporate all discussed revisions into the revised version.
> > >
> > > Best regards,
> > >
> > > The Authors

---

### Official Review · Reviewer_8V6U · 2026-03-09

**Soundness:** 3
**Presentation:** 3
**Significance:** 3
**Originality:** 2
**Overall Recommendation:** 4
**Confidence:** 2

**Summary:**

The paper proposes SimGFM, a discrete flow matching model for graph generation, and positions it within two existing DFM formulations: Campbell et al.’s formulation and Gat et al.’s VF-denoiser formulation. Different from earlier DFM-based graph generation methods that rely on the former, the proposed method is developed under the latter, which is presented as a simpler and more flexible alternative.

**Compliance With Llm Reviewing Policy:**

Affirmed.

**Final Justification:**

As stated in the rebuttal, the authors addressed most of my concerns.

**Key Questions For Authors:**

**Q1.** Why are the 10-step or 20-step results omitted for Guacamol and MOSES in Table 3? Does the low-step efficiency degrade as graph size and complexity increase?

**Q2.** Given that ESP directly samples the one-shot prediction in the last $k$ steps, does this sudden transition cause any issues with sample diversity or out-of-distribution generation compared to continuous flow updates?

**Q3.** How do you weight the node and edge components of the cross-entropy loss during training?

**Limitations:**

Yes.

**Strengths And Weaknesses:**

**Strengths**

1. The paper has a clear practical motivation. Reducing sampling steps for graph generation is worthwhile, and the paper focuses on a real pain point in current graph diffusion/flow methods.

**Weaknesses**

1. The proposed method seems to be combined from existing ingredients, including the VF-denoiser, a polynomial scheduler closely related to prior time-distortion approaches, and an endpoint switch to posterior sampling. These components are individually reasonable, but the overall contribution may be incremental.

2. Using sampling steps as an efficiency proxy is insufficient, since fewer steps do not necessarily mean faster or cheaper. The paper should report runtime and memory usage, ideally with baseline comparisons.

3. The *low tuning burden* claim feels overstated. SimGFM still relies on several hyperparameters (e.g., scheduler order $k$, ESP cutoff), and the ablations are too limited, especially lacking a real sensitivity analysis over $k$.

4. ESP drops the flow-matching updates in the last $k$ steps and samples directly from $p_{1|t}^{\theta}(\cdot|G_{t})$. While this avoids the negative-probability issue in Theorem 3.1, it introduces an abrupt endpoint “jump.” It would be helpful to quantify its impact by reporting FCD with/without ESP.

---

> ### Author Rebuttal · Authors · 2026-03-31
>
> Thank you for the careful review. We respond to each concern directly.
>
> **(W1) Why SimGFM is more than an incremental combination.** Our contribution is not stacking known components but showing that **their tightly coupled integration** yields a **minimal design principle** for **fast, high-fidelity** graph generation—governing **step allocation**, **probability-path shaping**, and **negative-probability handling**—behavior absent when each component is considered alone.
>
> Specifically: (1) we identify that **graph validity changes most sharply near the endpoint**, so sampling budget should concentrate there; (2) our **polynomial scheduler differs fundamentally from DeFoG’s time distortion**—it directly controls $\kappa_t$ and hence $\lambda_t$, providing a clearer probabilistic interpretation (Appendix C); (3) high-order scheduling inevitably causes negative probabilities (Theorem 3.1), and **ESP is a principled remedy for this instability**, not an ad hoc switch.
>
> This **coupled interaction** enables SimGFM to match 500–1000-step performance in only 10–50 steps.
>
>
> **(W2) Real efficiency: runtime and memory.** We measured wall-clock time (RTX 3090) and present results alongside timing:
>
> **Planar (40 graphs):**
>
> |Method (steps)|V.U.N.|Ratio|Time|
> |-|-:|-:|-:|
> |DeFoG (1000)|99.5|1.6|171.5s|
> |SimGFM (50)|99.5|1.8|6.5s|
> |SimGFM (200)|100.0|9.3|25.2s|
>
> **QM9 (10k graphs):**
>
> |Method (steps)|Valid|FCD|Time|
> |-|-:|-:|-:|
> |DeFoG (500)|99.3|0.12|438.6s|
> |SimGFM (10)|99.5|0.92|6.9s|
> |SimGFM (50)|99.7|0.13|32.7s|
> |SimGFM (200)|99.8|0.15|127.2s|
>
> **QM9H (10k graphs):**
>
> |Method (steps)|Valid|FCD|Time|
> |-|-:|-:|-:|
> |DeFoG (500)|98.0|0.05|2714.1s|
> |SimGFM (50)|98.4|0.10|215.8s|
> |SimGFM (200)|98.4|0.10|851.4s|
>
> Runtime scales linearly with steps; GPU memory is constant (`1.1/5.4 GB`), so fewer steps yield proportional savings. SimGFM-50 matches or exceeds DeFoG's best quality in **13-26x less time**: on Planar, identical V.U.N. `99.5` in `6.5s` vs `171.5s`; on QM9, Valid `99.7`/FCD `0.13` surpasses DeFoG-500 (`99.3`/`0.12`) in `32.7s` vs `438.6s`. On QM9H, SimGFM-50 (`215.8s`) achieves Valid `98.4`/FCD `0.10`, matching DeFoG-500 (`2714.1s`). Per step, vf-denoiser avoids DeFoG's rate-matrix construction and stabilization overhead.
>
> **(W3) Tuning burden and sensitivity over `k`.** SimGFM does **not** require an additional ESP hyperparameter search: Once the scheduler order $k$ and total step count are fixed, the ESP activation window is fully determined by the condition $\lambda_t > 1$ (Theorem 3.1). In contrast, DeFoG must tune both the guidance weight $\omega$ and the exploration weight $\eta$; in the official code this entails a grid over  `11 x 9 = 99` combinations, far more expensive than our one-dimensional search over `k`.
>
> Expanded `k` sensitivity analysis:
>
> |Setting|k=1|moderate k|high k|Best trade-off|
> |-|-|-|-|-|
> |QM9, 10 steps|Valid `75.8`, FCD `0.68`|`k=5`: Valid `99.3`, FCD `0.26`|`k=30`: Valid `99.5`, FCD `1.17`|moderate `k`|
> |Planar, 200 steps|V.U.N. `57.5`, Ratio `2.3`|`k=5`: `99.5`, `1.6`|`k=10`: `100.0`, `9.3`|moderate `k`|
> |Tree, 200 steps|V.U.N. `57.0`, Ratio `1.8`|`k=5`: `98.0`, `2.1`|`k=10`: `99.5`, `1.5`|higher `k`|
>
> The pattern is simple: too small `k` under-refines the endpoint; too large `k` over-concentrates probability mass after validity has saturated. Full per-dataset tables for all step regimes are in the [link](https://anonymous.4open.science/r/SimGFM-re-CC66/img/sensitivity_to_k.png).
>
> **(W4) Quantifying ESP's distributional impact.** We report FCD with/without ESP:
>
> |Setting|QM9 Valid/FCD|QM9H Valid/FCD|
> |-|-|-|
> |vf-denoiser, identity sched.|`98.3/0.10`|`97.7/0.10`|
> |+ESP, identity sched.|`99.3/0.10`|`95.6/0.10`|
> |vf-denoiser, our sched.|`99.6/0.11`|`97.7/0.10`|
> |+ESP, our sched.|`99.8/0.15`|`98.4/0.10`|
>
> ESP has minimal FCD impact (unchanged on QM9H; `0.11→0.15` on QM9). By comparison, increasing `k` from 5 to 30 raises QM9 FCD from `0.26` to `1.17` (10 steps), an order of magnitude larger.
>
> **(Q1) Low-step results on GuacaMol and MOSES.**
>
> |Dataset|10 steps|20 steps|
> |-|-|-|
> |GuacaMol|Valid `85.2`, FCD `55.3`|Valid `94.5`, FCD `73.2`|
> |MOSES|Valid `79.3`, FCD `1.14`|Valid `84.0`, FCD `1.11`|
>
> Performance degrades on larger molecules as expected, but 20-step SimGFM already surpasses DiGress and DeFoG-50 (GuacaMol FCD `73.2` vs `68.0`/`57.9`; MOSES FCD `1.11` vs `1.19`/`1.87`). The step requirement scales with graph complexity and is not specific to SimGFM.
>
> **(Q2)** As shown in W4, ESP has near-zero impact on FCD. Since FCD is sensitive to diversity loss, a collapse would cause a much larger degradation, which we do not observe. ESP acts only in the final steps, after global diversity is largely set, and it uses **posterior sampling** $p_{1|t}(x)$ rather than deterministic argmax, preserving stochasticity.
>
> **(Q3)** We weight edge and node cross-entropy losses by `5:1`, following DiGress.
>
> We hope these additions address the reviewer’s concerns and demonstrate SimGFM’s practical value.

---

> > ### Author Rebuttal · Reviewer_8V6U · 2026-04-02
> >
> > I thank the authors for their detailed response. Most of my concerns have been resolved, and I am willing to raise my score to 4 but with a low confidence. My remaining concerns are limited. One relates to novelty, which is partly a matter of subjective judgment, and the other is that the supplementary material does not provide implementation code, so the efficiency of SimGFM is difficult to verify.

---

> > > ### Author Response · Authors · 2026-04-03
> > >
> > > Dear Reviewer 8V6U,
> > >
> > > Thank you very much for your kind follow-up and for letting us know that you found our rebuttal helpful. We are glad that our response has addressed most of your concerns, and we fully appreciate your view that the assessment of novelty is, to some extent, subjective. We briefly clarify the remaining point below.
> > >
> > > **On code availability and efficiency verification.**
> > > We apologize for any confusion regarding the code. Our original submission **already included a complete anonymous code repository**: in **Line 969**, we stated: “Source code, datasets, and detailed instructions are available at https://anonymous.4open.science/r/SimGFM-2703 .” This repository contains the full training and sampling code and is sufficient to reproduce all reported results.
> > >
> > > According to the ICML 2026 Author Instructions, the anonymous repository associated with the original submission must remain on an unchanged branch after the submission deadline, so we were unable to modify it during the rebuttal period. For this reason, we provided a separate rebuttal-period link (https://anonymous.4open.science/r/SimGFM-re-CC66), which was used only to share additional figures and clarifications.
> > >
> > > We apologize for the inconvenience. In future submissions, we will ensure that the code link is placed more prominently in the main text and also included in the supplementary material.
> > >
> > > Thank you once again for your thoughtful feedback and for your careful, conscientious review.
> > >
> > > Best regards,
> > >
> > > The Authors

---

### Official Review · Reviewer_VcGC · 2026-03-11

**Soundness:** 3
**Presentation:** 3
**Significance:** 3
**Originality:** 3
**Overall Recommendation:** 4
**Confidence:** 2

**Summary:**

This paper proposes SimGFM, a simplified discrete flow matching (DFM) framework for graph generation. The authors argue that existing graph DFM methods introduce excessive complexity through task-specific heuristics that violate the continuity equation and expand the hyperparameter space. SimGFM makes three design choices: (1) adopting the vf-denoiser formulation for an implementation-friendly rate matrix, (2) employing a polynomial endpoint-focused scheduler motivated by the observation that valid graph structures emerge near t→1, and (3) introducing Endpoint Safety Projection (ESP) to handle numerical invalidity in the tail window.

**Compliance With Llm Reviewing Policy:**

Affirmed.

**Final Justification:**

I appreciate the authors' response. After the rebuttal discussion, as my concerns have been adequately addressed, I maintain my positive rating.

**Key Questions For Authors:**

Please see weaknesses.

**Limitations:**

Please see weaknesses.

**Strengths And Weaknesses:**

**Strengths**
1. The paper shows clear evidence (through graphs and experiments) that focusing on the final steps of the generation process makes sense—especially when using uniform denoising. This gives a solid reason for their design choices.
2. The method achieves results that are just as good as existing methods, but in far fewer steps.
3. The authors test their method on many different types of graphs—molecules, social networks, grids, etc.—and include conditional generation too. They also run ablation studies and report error margins, making the results more comprehensive.

**Weaknesses**
1. The core ideas (like the denoiser formula, the time scheduler, and the ESP trick) are borrowed from recent papers. The main contribution is putting them together, not inventing something fundamentally new.
2. Some experimental details are missing or inconsistent: 1) It’s hard to compare methods fairly because step counts aren’t always matched. 2) The key parameter k isn’t clearly reported for each experiment, making replication tricky. 3) In one case (on Planar at 200 steps (Table 1)), the method gets perfect validity but poor diversity (low “Ratio”), which isn’t explained.
3. Ablation studies are expected to be further enriched. The authors only compare their scheduler against a basic baseline (k=1), so it’s unclear how sensitive performance is to k. They also don’t explore how choices like the starting noise distribution affect results, even though these vary across datasets.

---

> ### Author Rebuttal · Authors · 2026-03-31
>
> Thank you for your constructive feedback and for recognizing the clear motivation, strong efficiency, and broad evaluation of our work. We address your concerns below.
>
> **(W1) Novelty beyond combining existing ingredients.** We agree that each component has theoretical roots, but SimGFM is **not** a simple stacking of known tricks. The key observation driving the design—that graph validity changes most sharply near $t \to 1$ in generation trajectories (Section 3.2)—is an **original empirical finding** of this paper that had not been systematically analyzed or exploited before. The choice of vf-denoiser, polynomial scheduler, and ESP are motivated by this observation.
>
> ESP is not an ad hoc patch but a theoretically grounded scheduler correction. Under the vf-denoiser, the Euler update takes the form
>
> $p_{t+h}(x) = \lambda_t  p_{1|t}(x) + (1-\lambda_t)  \delta_t(x), \quad \lambda_t = \frac{h\dot{\kappa}_t}{1-\kappa_t},$
>
> which yields valid probabilities only when $\lambda_t \leq 1$. ESP simply clamps $\lambda_t' = \min(\lambda_t, 1)$, which is equivalent to a corrected scheduler $\kappa_t'$ that still satisfies all DFM conditions ($\kappa_0=0$, $\kappa_1=1$, monotonicity, differentiability)—see our response to Reviewer TfNh (W1) for the full derivation.
>
> Thus the three components are **internally coupled**: the polynomial scheduler concentrates updates near the endpoint, improving validity but making the Theorem 3.1 instability more likely; ESP resolves exactly this instability. Ablations confirm that removing any single component degrades performance.
>
>
> **(W2-1) Fair comparison with matched step counts.** We appreciate this point and added step-matched results:
>
> |Dataset|Matched steps|SimGFM result|
> |-|-:|-|
> |Planar|1000|V.U.N. `100.0±0.0`, Ratio `21.0±1.5` (k=10; note: **k=2 gives Ratio `2.2±0.4`**, see W2-3)|
> |Tree|1000|V.U.N. `100.0±0.0`, Ratio `2.1±0.8`|
> |SBM|1000|V.U.N. `93.0±1.9`, Ratio `8.2±3.0`|
> |QM9|500|Valid `99.8±0.0`, FCD `0.31±0.0`|
> |QM9H|500|Valid `98.6±0.0`, FCD `0.10±0.0`|
> |GuacaMol|500|V.U.N. `98.6`, FCD `79.7`|
> |MOSES|500|Valid `91.1`, FCD `1.19`|
>
> **(W2-2) Reporting `k` for each experiment.** The selected scheduler order `k` for 200/50/10-step settings: `QM9:20`, `QM9H:10`, `GuacaMol:10`, `MOSES:10`, `Tree:10`, `Planar:10`, `SBM:3`, `TLS/Ego-Small/Community-small/Grid:10`. We will include these in the revision for reproducibility.
>
> **(W2-3) Why does Planar at 200 steps show perfect validity but worse Ratio?** As analyzed in Section 3.2, sufficient iteration near the endpoint ($t \to 1$) helps refine invalid structures into valid ones, improving validity. However, once validity is already near saturation, continuing dense endpoint updates can hurt diversity: the model prediction $p_{1|t}^{\theta}$ receives dominant weight in the update and tends to concentrate probability mass on frequent structures, gradually overwriting low-frequency but still valid ones. Since Ratio measures how well the generated structural distribution matches the training distribution, losing those valid low-frequency modes naturally increases Ratio. We ran a dedicated ablation at 200 and 1000 steps:
>
> |Planar|k=1|k=2|k=5|k=10|k=20|k=30|
> |-|-|-|-|-|-|-|
> |200-step V.U.N. (%)|57.5|97.0|99.5|100.0|100.0|99.5|
> |200-step Ratio|2.3|1.6|1.6|9.3|8.2|15.4|
> |1000-step V.U.N. (%)|93.5|100.0|100.0|100.0|-|-|
> |1000-step Ratio|1.4|2.2|19.9|21.0|-|-|
>
> These results support this interpretation. At 200 steps, `k=5` (`99.5`, `1.6`) greatly improves Ratio over `k=10` (`100.0`, `9.3`) with almost no validity loss; at 1000 steps, `k=2` (`100.0`, `2.2`) is likewise better than larger `k`. QM9 at 10 steps shows the same pattern: as `k` increases, validity gains saturate while FCD degrades quickly. We also note that MMD-based Ratio can fluctuate under small-sample evaluation \[1, 2\].
>
> **(W3) Sensitivity over `k` and source distribution.** We summarize the key findings here and refer to our response to Reviewer 8V6U (W3) for the full table. On QM9 (10 steps), `k=1` yields only `75.8%` validity, moderate `k=5` achieves `99.3%`/FCD `0.26`, while `k=30` over-concentrates and degrades FCD to `1.17`. The pattern is consistent across datasets: too small `k` under-allocates endpoint updates, while too large `k` over-concentrates them after validity saturates; a moderate `k` gives the best trade-off. SimGFM reduces DeFoG's 99-combination search ($\omega \times \eta$) to a single `k`, with a clear and consistent tuning pattern. Our default source prior is Marginal (Absorbing only for SBM). Across 6 QM9 priors at 10 steps ([figure](https://anonymous.4open.science/r/SimGFM-re-CC66/img/qm9_sensitivity_to_source_distribution.png)), Uniform, Marginal, Node-Marginal, and Edge-Marginal all exceed `99%` validity, while Absorbing and Masking underperform.
>
> ---
>
> [1] Evaluation Metrics for Graph Generative Models: Problems, Pitfalls, and Practical Solutions. ICLR, 2022.
>
> [2] PolyGraph Discrepancy: A Classifier-Based Metric for Graph Generation. ICLR, 2026.

---

> > ### Author Rebuttal · Reviewer_VcGC · 2026-04-03
> >
> > Thank you for the response. I hope to see these improvements in the revised version. My positive score remains unchanged.

---

> > > ### Author Response · Authors · 2026-04-03
> > >
> > > Dear Reviewer VcGC,
> > >
> > > We sincerely appreciate your positive follow-up and your confirmation that our rebuttal has addressed your concerns. We will carefully incorporate the discussed improvements in the revised version.
> > >
> > > Best regards,
> > >
> > > The Authors

---

### Decision · Program_Chairs · 2026-04-30

**Decision:**

Accept (regular)

**Comment:**

Reviewers agreed that the paper has a clear motivation, a well-structured presentation, and strong empirical gains in graph generation efficiency. They also raised concerns about the paper's novelty and experimental completeness: the method was seen as largely combining existing DFM ingredients, some step-count and hyperparameter reporting was initially unclear, and the ablation study needed to better quantify the roles of the scheduler, $k$ and ESP. The rebuttal addressed these points with matched-step runtime/memory results, detailed $k$-sensitivity analysis, and additional explanation of the endpoint behavior and ESP, which substantially strengthened the paper’s technical soundness. Therefore, I tend to accept this paper.